# GNeSF: Generalizable Neural Semantic Fields

**Hanlin Chen   Chen Li   Mengqi Guo   Zhiwen Yan   Gim Hee Lee**
Department of Computer Science, National University of Singapore
{hanlin.chen, gimhee.lee}@comp.nus.edu.sg

## Abstract

3D scene segmentation based on neural implicit representation has emerged recently with the advantage of training only on 2D supervision. However, existing approaches still requires expensive per-scene optimization that prohibits generalization to novel scenes during inference. To circumvent this problem, we introduce a *generalizable* 3D segmentation framework based on implicit representation. Specifically, our framework takes in multi-view image features and semantic maps as the inputs instead of only spatial information to avoid overfitting to scene-specific geometric and semantic information. We propose a novel soft voting mechanism to aggregate the 2D semantic information from different views for each 3D point. In addition to the image features, view difference information is also encoded in our framework to predict the voting scores. Intuitively, this allows the semantic information from nearby views to contribute more compared to distant ones. Furthermore, a visibility module is also designed to detect and filter out detrimental information from occluded views. Due to the generalizability of our proposed method, we can synthesize semantic maps or conduct 3D semantic segmentation for novel scenes with solely 2D semantic supervision. Experimental results show that our approach achieves comparable performance with scene-specific approaches. More importantly, our approach can even outperform existing strong supervision-based approaches with only 2D annotations. Our source code is available at: https://github.com/HLinChen/GNeSF.

## 1   Introduction

Understanding high-level semantics of 3D scenes in digital images and videos is a fundamental research in computer vision. Several extensively studied tasks such as scene classification [22], object detection [34], and semantic segmentation [10] facilitate the extraction of semantic descriptions from RGB and other sensor data. These tasks form the foundation for applications such as visual navigation [4] and robotic interaction [2].

Several commonly used 3D representations include 3D point clouds [11, 32], voxel grids [42], and polygonal meshes [28, 18]. A direct learning of semantics on these 3D representations require large amounts of well-annotated 3D ground truths for training. Unfortunately, these 3D ground truths are often much more laborious and expensive to obtain compared to its 2D counterparts. Recently, Neural Radiance Field (NeRF) [30, 52] has emerged as a new 3D representation, which can capture intricate geometric details [41] using solely RGB images. Many researchers utilize this 3D representation to propose the concept of Neural Semantic Field [55, 17, 3], where semantics is assigned to each 3D point. However, these existing works on neural semantic fields inherited the per-scene optimization limitation of the vanilla NeRF and thus cannot generalize across novel scenes. As a result, the practicality of the approaches for semantic segmentation is severely restricted.

In view of the limitations in existing neural semantic field works, this paper addresses the challenging task of generalizable neural semantic fields. The objective is to construct a generalizable semantic field capable of producing high-quality semantic rendering from novel views and 3D semantic

37th Conference on Neural Information Processing Systems (NeurIPS 2023).

segmentation using input images from unseen scenes. The generalization ability offers numerous interactive applications that provide valuable feedback to users during 3D capture without the requirement for retraining. Recently, NeSF [44] claims the ability to generate 3D semantic fields for novel scenes by utilizing 3D UNet [36] to extract semantic features from a density field obtained from a pre-trained NeRF model. However, this method lacks efficiency as it requires training a separate NeRF model for each scene before segmenting the 3D scene. Furthermore, it lacks sufficient generalizability to new scenes, thus making it suitable only for offline applications. Recently, several researchers propose generalizable NeRF methods [6, 45, 53] that demonstrate effective generalization across scenes for reconstructing radiance fields without the need for laborious per-scene optimization. A naive approach for generalizable Neural Semantic Field (NeSF) is to incorporate an additional semantic head to the generalizable NeRF framework. However, this method does not perform well for complex and realistic indoor scenes, as demonstrated in Tab. 1 and [20]. One possible reason is that semantics represents high-level information, and a simple head without direct 3D semantic supervision is insufficient for generating 3D semantic fields.

Inspired by image-based rendering (IBR) [9, 45], we propose a soft voting scheme (shown in Fig. 2a) to aggregate the 2D semantic information from different views for each 3D point for learning a Generalizable Neural Semantic Field (GNeSF). Specifically, we use a pre-trained 2D semantic segmentation model to generate 2D semantic maps for the source views and then leverage the warping, resampling, and/or blending operations of IBR to infer the semantics of a sample 3D point. Considering the unbounded nature and the lack of meaning for logit values across multiple views, we utilize a probability distribution instead of semantic logits to represent semantics across views. Our framework utilizes multi-view image features instead of position information as input to further enhance the generalizability of our model and prevent overfitting to scene-specific geometric and semantic information. In addition to the extracted image features, our framework incorporates view difference information to include the prior of higher significance for nearby views when generating the voting scores for blending source views. We then introduce a soft voting scheme to compute the semantics of each sample point as the weighted combination of the projected semantics from the source views.

Geometric information is predicted to perform volumetric rendering after obtaining semantics of 3D points along sampled ray. Specifically, we first build a feature volume by projecting each vertex of the volume to latent 2D features and aggregating these 2D features to get a density 3D feature. This feature volume is then fed into a 3D neural network to obtain a geometry encoding volume. For a sampled 3D point, we use an MLP network to infer the specific geometric representations, e.g., density or Signed Distance Function (SDF), from the interpolated features of the geometry encoding volume.

Weights of points along the ray are obtained from the predicted geometric representations. A final semantic value is computed for each ray through volume rendering by weighted summing semantics alone the ray. Due to the generalizability of our proposed method, we can synthesize semantic maps or conduct 3D semantic segmentation for novel scenes with solely 2D semantic supervision. To facilitate 3D semantic segmentation, we design a visibility module (shown in Fig. 2b) that identifies and removes occluded views for each spatial point. The remaining semantics from the visible views are then utilized to vote on the semantics of the respective point. In summary, our contributions include:

- We are the first to tackle the task of *generalizable* neural semantic field. Our proposed GNeSF is capable of inferring novel view synthesis of semantic maps or 3D semantic segmentation of novel scenes without retraining. Furthermore, our method depends solely on 2D semantic supervision during training.

- We propose a novel soft voting scheme that determines the semantics for each 3D point by considering the 2D semantics from multiple source views. A visibility module is also designed to detect and filter out detrimental information from occluded views.

- Our proposed GNeSF enables synthesis of semantic maps for both novel views and novel scenes, and achieves comparable performance to per-scene optimized methods. The results on ScanNet and Replica demonstrate comparable or superior performance compared to existing methods utilizing strong 3D supervision.

## 2 Related Work

### 2.1 2D & 3D Semantic Segmentation

Traditionally, semantic segmentation has been treated as a per-pixel classification task, where FCN-based architectures [27] assign category labels to individual pixels independently. Subsequent methods have acknowledged the importance of context in achieving accurate per-pixel classification and have concentrated on developing specialized context modules [7, 8, 54] or incorporating self-attention variations [39, 46, 10]. In our work, we utilize the state-of-the-art Mask2Former [10] with a Swin-Transformer [26] backbone as our pre-trained 2D semantic segmenter. 3D semantic segmentation also has been investigated with pre-computed 3D structures such as voxel grids or point clouds [19, 32, 33, 37, 12, 14, 35], and simultaneous segmentation and 3D reconstruction from 2D images [31, 29]. Unlike these methods, our GNeSF aims to segment a dense 3D representation using only 2D inputs and semantic supervision without the need for 3D semantic annotations.

### 2.2 Neural Semantic Fields

Semantic-NeRF [55] is first to introduce the integration of semantics into NeRF. It showcases the fusion of noisy 2D semantic segmentations into a coherent volumetric model. This integration enhanced the accuracy of the model and enabled the synthesis of novel views with semantic masks. Subsequently, numerous studies have built upon this concept. For example, [17, 23, 21] incorporate instance modeling, and [43] encode abstract visual features that allow for post hoc derivation of semantic segmentation. NeRF-SOS [16] combines object segmentation and neural radiance fields to segment objects in complex real-world scenes using self-supervised learning. [50] proposes an object-compositional neural radiance field. RFP [25] designs an unsupervised approach to segment objects in 3D during reconstruction using only unlabeled multi-view images of a scene. Panoptic NeRF [17] and DM-NeRF [3] are specifically designed for panoptic radiance fields for tasks such as label transfer and scene editing, respectively. However, these methods require scene-specific optimization and thus not generalizable to unseen scenes. Similar to our objective, NeSF [44] addresses the challenge of generalization to unseen scenes. Nonetheless, NeSF relies on multiple pre-trained and scene-specific optimized NeRF models for density field generation and segmentation which restricts its overall generalizability. In contrast, our GNeSF is able to infer on novel scenes directly without any further optimization.

### 2.3 Generalizable Neural Implicit Representation

Significant advancements have been achieved in the field of neural implicit representation [30, 48, 52]. However, these approaches heavily depend on computationally expensive per-scene optimization. To address this limitation, several new neural implicit representation methods emphasizing on generalization has emerged. These methods [6, 45, 49, 51, 5] aim to learn a representation of the radiance field from a provided set of images of novel scenes. This enables the synthesis of novel views without requiring per-scene optimization. PixelNeRF [51] and IBRNet [45] utilize volume rendering techniques to generate novel view images by employing warped features from nearby reference images. In addition to novel synthesis, MonoNeuralFusion [56] introduces a novel neural implicit scene representation with volume rendering for high-fidelity generalizable 3D scene reconstruction from monocular videos. Inspired by these generalizable methods, we takes in multi-view image feature as the input instead of the position information to enhance the generalizability of our model. Semantic knowledge obtained from 2D vision is then aggregated to infer 3D semantics by the proposed soft voting scheme.

## 3 Our GNeSF: Generalizable Neural Semantic Fields

We propose GNeSF to achieve generalizable semantic segmentation on neural implicit representation. As illustrated in Fig. 1, our GNeSF consists of three main components: 1) Feature extraction and semantic prediction from multiple source views; 2) Geometry and semantics prediction for sampled 3D points; 3) Semantic map prediction with volume rendering.

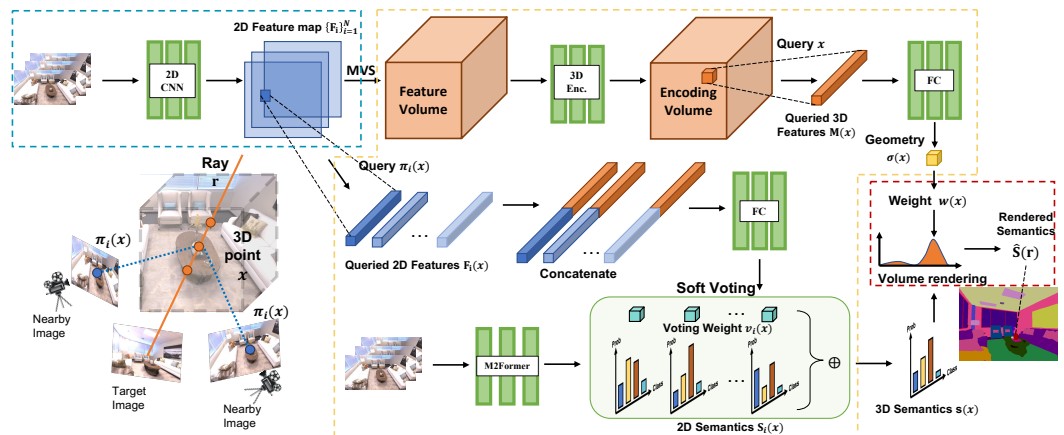

Figure 1: The GNeSF framework comprises three key components: 1) Extraction of features and semantic prediction from multiple source views, shown in the blue dashed box; 2) Prediction of geometry and semantics for sampled 3D points, shown in the yellow dashed box; 3) Prediction of semantic maps using volume rendering, shown in the red dashed box. Here, "MVS" means using Multi-View Stereo to build the feature volume.

## 3.1 Source-View Image Features and Semantics Map

Inspired by existing generalizable neural implicit representations [6, 45], our framework leverages image features from source-view images for generalization to unseen scenes during inference. We extract an image feature map $\mathbf{F}_i \in \mathbb{R}^{H_i \times W_i \times d}$ for each source image $\mathbf{I}_i \in [0,1]^{H_i \times W_i \times 3}$ with a U-Net-based convolutional neural network, where $H_i \times W_i$ and $d$ denotes the dimensions of the image and the feature maps, respectively. Concurrently, we use a pre-trained mask2former [10] to predict the semantic maps $\mathbf{S}_i \in \mathbb{R}^{H_i \times W_i \times c}$ of each source image $\mathbf{I}_i$, where $c$ represents the number of categories.

## 3.2 3D Point Geometry and Semantics Predictions

Using the source-view image features and semantics maps, we predict the geometry $\sigma(\mathbf{x})$ and semantics $\mathbf{s}(\mathbf{x})$ for each sampled 3D point $\mathbf{x} \in \mathbb{R}^3$. We first build a geometry encoding volume $\mathbf{M}$ based on the image feature map $\mathbf{F}_i$ and then predict the geometry $\sigma(\mathbf{x})$ using an MLP. Concurrently, we design a soft voting scheme (shown in Fig. 2a) based on the 2D semantic observations $\mathbf{S}_i \in \mathbb{R}^{H_i \times W_i \times c}$ to predict the semantics $\mathbf{s}(\mathbf{x})$.

### 3.2.1 Geometry Encoding Volume

To construct the geometry encoding volume M, we project each vertex $\mathbf{k} \in \mathbb{R}^3$ of the volume to the image feature maps $\mathbf{F}_i$ and obtain its image features $\mathbf{F}_i(\pi_i(\mathbf{k}))$ by interpolation, where $\pi_i(\mathbf{k})$ denotes the projected pixel location of $\mathbf{k}$ on the feature map $\mathbf{F}_i$. For simplicity, we abbreviate $\mathbf{F}_i(\pi_i(\mathbf{k}))$ as $\mathbf{F}_i(\mathbf{k})$. We then compute the mean $\mu \in \mathbb{R}^d$ and variance $\mathbf{v} \in \mathbb{R}^d$ of the projected image features $\{\mathbf{F}_i(\mathbf{k})\}_{i=1}^N$ from the $N$ source views to capture the global information. The mean $\mu$ and variance $\mathbf{v}$ are concatenated to build a feature volume $\mathbf{B}$. Intuitively, the mean image feature provides cues for the appearance information, and the variance for geometry reasoning. Finally, we use a 3D neural network $H_g$ to aggregate the feature volume $\mathbf{B}$ to obtain the geometry encoding volume $\mathbf{M}$. Formally, the whole process can be expressed by:

$$\mu(\mathbf{k}) = \text{Mean}(\{\mathbf{F}_i(\mathbf{k})\}_{i=1}^N), \quad \mathbf{v}(\mathbf{k}) = \text{Var}(\{\mathbf{F}_i(\mathbf{k})\}_{i=1}^N), \tag{1a}$$

$$\mathbf{B}(\mathbf{k}) = [\mu(\mathbf{k}), \mathbf{v}(\mathbf{k})], \quad \mathbf{M} = H_g(\mathbf{B}), \tag{1b}$$

where $[\cdot, \cdot]$ represents feature concatenation, and $\text{Mean}(\cdot)$ and $\text{Var}(\cdot)$ are the averaging and variance operations. For a sampled 3D point $\mathbf{x}$, we use an MLP network $f_\theta$ to predict the geometry $\sigma(\mathbf{x})$ based on the interpolated features of the geometry encoding volume $\mathbf{M}(\mathbf{x})$, *i.e.*:

$$\sigma(\mathbf{x}) = f_\theta(\mathbf{M}(\mathbf{x})). \tag{2}$$

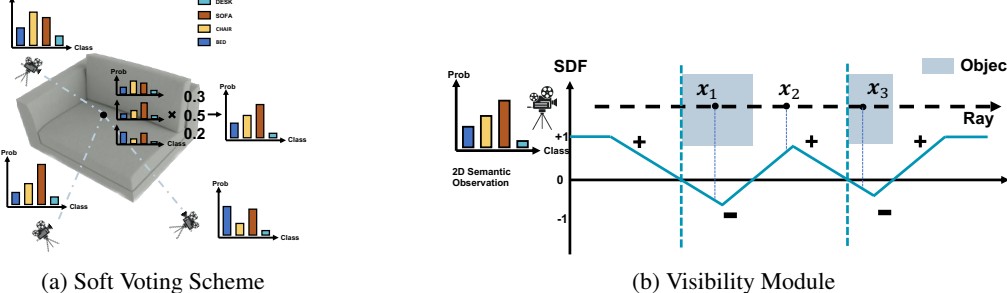

|     |     |
| --- | --- |
| (a) Soft Voting Scheme | (b) Visibility Module |

Figure 2: (a) The soft voting scheme aggregates the 2D semantic information from different views to infer semantics of each 3D point. (b) The visibility module detects and eliminates occluded views for each spatial point. For example, $\mathbf{x}_1$ is visible in the 2D semantic observation because the SDF values along the ray from the source view to the target point transits from positive to negative. On the other hand, $\mathbf{x}_2$ and $\mathbf{x}_3$ are occluded in the 2D semantic observation because the value changes from positive to negative and then back to positive.

Note that geometry $\sigma(\mathbf{x})$, e.g., volume density or Signed Distance Function (SDF) is represented differently in different neural implicit representations. More details are provided in Sec. 4.

### 3.2.2 Semantics Soft Voting

Given a query point location $\mathbf{x} \in \mathbb{R}^3$ along a ray $\mathbf{r} \in \mathbb{R}^3$, we project $\mathbf{x}$ onto $N$ nearby source views using the respective camera parameters. Image features $\{\mathbf{F}_i(\mathbf{x})\}_{i=1}^N \in \mathbb{R}^d$ and semantics maps $\{\mathbf{S}_i(\mathbf{x})\}_{i=1}^N \in [0,1]^k$ are then sampled at the projected pixel locations over the $N$ source views through bilinear interpolation on the image feature $\mathbf{F}_i$ and semantic maps $\mathbf{S}_i$, respectively. We propose a soft voting scheme to predict the semantics $\mathbf{s}(\mathbf{x})$ of a sampled 3D point $\mathbf{x}$ based on the 2D projected image features $\{\mathbf{F}_i(\mathbf{x})\}_{i=1}^N$ and semantics maps $\{\mathbf{S}_i(\mathbf{x})\}_{i=1}^N$. Specifically, the 2D projected image features $\{\mathbf{F}_i(\mathbf{x})\}_{i=1}^N$ are used to predict voting weights $\{v_i\}_{i=1}^N$ for the corresponding semantics maps $\{\mathbf{S}_i(\mathbf{x})\}_{i=1}^N$ in the source views.

We first calculate the mean and variance of the image features $\{\mathbf{F}_i(\mathbf{x})\}_{i=1}^N$ from the $N$ source views to capture the global consistency information. Each image feature $\mathbf{F}_i(\mathbf{x})$ is concatenated with the mean and variance together, and then fed into a tiny MLP network to generate a new feature $\mathbf{F}'_i(\mathbf{x})$. Subsequently, we feed the new feature $\mathbf{F}'_i(\mathbf{x})$, the displacement $\Delta\mathbf{d}_i = \mathbf{x} - \mathbf{o}_i$ between the 3D point $\mathbf{x}$ and the camera origin $\mathbf{o}_i$ of the corresponding source view, and the trilinearly interpolated volume encoding feature $\mathbf{M}(\mathbf{x})$ into an MLP network $f_v$ to generate voting weights $v_i(\mathbf{x})$. In contrast to view-dependent color, we use only displacement for semantics since it remains consistent regardless of the viewing direction.

Furthermore, we encode the displacement information $\Delta\mathbf{d}_i = \mathbf{x} - \mathbf{o}_i$ for voting weights prediction since nearby views should contribute more compared to the distant ones. The voting weight $v_i(\mathbf{x})$ is predicted as:

$$v_i(\mathbf{x}) = f_v(\mathbf{F}'_i(\mathbf{x}), \mathbf{M}(\mathbf{x}), \Delta\mathbf{d}_i). \tag{3}$$

The final semantics of the 3D point is obtained through a soft voting mechanism, given by:

$$\mathbf{s}(\mathbf{x}) = \sum_{i=1}^N \left( \mathbf{S}_i(\mathbf{x}) \cdot \exp\left(v_i(\mathbf{x})\right) / \sum_{j=1}^N \exp\left(v_j(\mathbf{x})\right) \right), \tag{4}$$

where $\exp(.)$ represents the exponential function.

**Remarks.** An alternative approach would be to directly regress $\mathbf{s}(\mathbf{x})$ instead of predicting the voting weights. Compared to this alternative approach, predicting the voting weights makes it easier for the network to leverage the source-view semantics maps which are easier to obtain without direct 3D supervision. Our conjecture is verified experimentally by the degraded performance when we directly regress the semantics.

### 3.3 Semantics Rendering and Training

**Rendering.** The method presented in the previous section enables the computation of semantics $\mathbf{s}(\mathbf{x})$ and geometry $\sigma(\mathbf{x})$, e.g., volume density or SDF in the continuous 3D space. The volume rendering of the semantics $\hat{\mathbf{S}}(\mathbf{r})$ along a ray $\mathbf{r}$ traversing the scene is approximated by the weighted sum of the semantics from the $M$ samples along the ray, *i.e.*:

$$\hat{\mathbf{S}}(\mathbf{r}) = \sum_{m=1}^{M} w(\mathbf{x}_m) \cdot \mathbf{s}(\mathbf{x}_m), \tag{5}$$

where the samples from $1$ to $M$ are sorted in ascending order based on their depth values. $\mathbf{x}_m$ represents the query point of the $m$-th sample along the ray $\mathbf{r}$. $\mathbf{s}(\mathbf{x}_m)$ and $w(\mathbf{x}_m)$ represent the semantics and the corresponding weight of the query sample $\mathbf{x}_m$, respectively. The computation of the weight $w(\mathbf{x}_m)$ depends on the task of view synthesis or 3D segmentation (*c.f.* Sec. 4 for details).

**Training objective.** To train the model, we minimize the cross-entropy error between the rendered semantics $\hat{\mathbf{S}}(\mathbf{r})$ and the corresponding ground truth pixel semantics $\mathbb{S}(\mathbf{r})$.:

$$\mathcal{L}_s = -\frac{1}{|R|} \sum_{\mathbf{r} \in R} \mathbb{S}(\mathbf{r}) \log \hat{\mathbf{S}}(\mathbf{r}), \tag{6}$$

where $R$ represents the set of rays included in each training batch.

## 4 Semantic View Synthesis and 3D Segmentation

Our proposed generalizable neural semantic field can be applied to the tasks of semantic view synthesis and 3D semantic segmentation.

### 4.1 Semantic View Synthesis

Semantic view synthesis aims to render the semantic maps for novel views. A naive approach for synthesizing novel semantic views first uses a generalizable NeRF to generate images of the novel views and then followed by applying 2D semantic segmentation to these synthesized images using a separate 2D semantic segmentor. In contrast to this two-stage method, our approach is a one-step synthesis that is more efficient and streamlined. In semantic view synthesis, we use the predicted geometry $\sigma(\mathbf{x}_m)$ from Eq. 2 as the volume density. We then follow NeRF [30] to compute the weight $w(\mathbf{x}_m)$ used in Eq. 5 by:

$$w(\mathbf{x}_m) = T(\mathbf{x}_m) \left(1 - \exp(-\sigma(\mathbf{x}_m))\right), \quad \text{where } T(\mathbf{x}_m) = \exp(-\sum_{j=1}^{m-1} \sigma(\mathbf{x}_m)) \tag{7}$$

is the accumulated transmittance along the ray. Once the weights $w(\mathbf{x}_m)$ are computed, we proceed to construct and train a generalizable neural semantic field based on the approach proposed in Sec. 3.3 for rendering semantic views.

### 4.2 3D Semantic Segmentation

The goal of 3D semantic segmentation is to predict the semantic of each sample point in the 3D space. Consequently, the rendering of the semantic map is dependent on high-quality surfaces. To this end, we predict the geometry $\sigma(\mathbf{x}_m)$ as a signed distance field (SDF) based on Eq. 2. Following [1], we then compute the weights $w(\mathbf{x}_m)$ by:

$$w(\mathbf{x}_m) = \text{Sigmoid}\left(\frac{\sigma(\mathbf{x}_m)}{tr}\right) \cdot \text{Sigmoid}\left(-\frac{\sigma(\mathbf{x}_m)}{tr}\right), \tag{8}$$

where $tr$ represents the truncation distance. We set the weights of samples beyond the first truncation region to zero to account for possible multiple intersections. Subsequently, the normalized weight $w(\mathbf{x})$ used in Eq. 5 is calculated as:

$$w(\mathbf{x}_m) = \frac{w(\mathbf{x}_m)}{\sum_{j=1}^{M} w(\mathbf{x}_j)}. \tag{9}$$

In this scheme, the highest integration weight is assigned to surface points, while the weights for the points that are farther away are reduced.

**Visibility module.**  While most nearby views offer valuable semantic observations for a 3D point $\mathbf{x}$, there exists some views that contribute negatively due to occlusion. The most straightforward approach to address occlusion is to render the depth map in the same manner as color or semantics. However, this method incurs additional computational cost. To address occlusion efficiently, we leverage the property of signed distance function (SDF): the SDF value is positive in front of the surface and negative behind it. This implies that if a source view can observe a target 3D point, the SDF values of the points along the ray transit from positive to negative at the origin of the source view to the target point (*c.f.* $\mathbf{x}_1$ in Fig.2b). However, in the presence of occlusion, the value changes from positive to negative and then back to positive (*c.f.* $\mathbf{x}_2$ and $\mathbf{x}_3$ in Fig.2b). We use this property to mask out occluded views and mitigate the influence of occlusion.

# 5  Experiments

We evaluate our method on the tasks of semantic view synthesis in Sec. 5.1.1 and 3D semantic segmentation in Sec. 5.1.2. Additionally, we validate the effectiveness of the proposed modules in Sec. 5.2.

**Dataset and Metrics.**  We perform the experiments on two indoor datasets: ScanNet (V2) [13] and Replica [38]. The ScanNet dataset contains $1,613$ indoor scenes with ground-truth camera poses, surface reconstructions, and semantic segmentation labels. We follow [31] and [40] on the three training/validation/test splits commonly used in previous works. The 2D semantic segmentor Mask2Former is pre-trained on the train split of ScanNet without any additional amendment. Furthermore, we follow the 20-class semantic segmentation task defined in the original ScanNet benchmark. In contrast to NeRF-based methods [53, 56] which only use the training subset of ScanNet to train and test their methods, we train and test our method on the complete dataset. Replica consists of 16 high-quality 3D scenes of apartments, rooms, and offices. We use the same 8 scenes as in [55]. While Replica originally features 100 classes, we rearranged them into 19 frequently observed semantic classes. We use the model trained on ScanNet to perform the validation on Replica. For the evaluation metrics, we use the mean of class-wise intersection over union (mIoU) and the average pixel accuracy (mAcc). For 3D semantic segmentation, we follow the evaluation procedure of Atlas [31].

Table 1: Quantitative comparison on semantic view synthesis from the validation set of ScanNet. 'Ours-100' means the model trained on 100 scenes and 'Ours-1000' trained on full training set of ScanNet. Semantic-Ray is also trained on full training set of ScanNet. 'Predict Directly' represents the method that incorporates a semantic head into the generalizable NeRF framework.

| | Method | Novel View | | Novel Scene | |
|---|---|---|---|---|---|
| | | mIoU (%) | mAcc (%) | mIoU (%) | mAcc (%) |
| Per-scene optimized | Semantic-NeRF [55] | **96.8** | **98.2** | - | - |
| | DM-NeRF [3] | 93.5 | 96.1 | - | - |
| | PNF [23] | 94.1 | 97.2 | - | - |
| Gneralizable | Mask2Former [10] | 80.9 | 90.7 | 64.6 | 77.9 |
| | Predict Directly | 50.2 | 62.3 | 39.1 | 54.2 |
| | Semantic-Ray [24] | - | - | 56.0 | - |
| | Ours-100 | **93.3** | **96.3** | 47 | 60.8 |
| | Ours-1000 | 87.8 | 93.3 | **71.6** | **82.3** |

## 5.1  Comparisons to Baselines

### 5.1.1  Semantic View Synthesis

We compare with state-of-the-art per-scene optimized neural semantic field methods: Semantic-NeRF [55], Panoptic Neural Fields (PNF) [23] and DM-NeRF [3]. For each per-scene optimized method, we train a NeRF model on each scene from the train set of ScanNet, and then compute the 2D mIoU

across these scenes. Note that although PNF and DM-NeRF aim to predict panoptic segmentation, they can still predict semantic segmentation. We train our method for different number of scenes, including 100 scenes (denoted as Ours-100) and full training set (denoted as Ours-1000), to show the capacity of our model over different scenarios. For generalizable methods, we compare with Mask2Former, Sematic-Ray [24] and the method adding a semantic head to a generalizable NeRF (denoted as Predict Directly) in the same training and testing setting. For Mask2Former, IBRNet [45] is first used to render color images on novel views, and then Mask2Former is used to predict semantic results on rendered images. For Predict Directly, we use IBRNet as its generalizable NeRF framework.

As shown in Tab. 1, we outperform the baselines across ScanNet on semantic segmentation tasks. Compared with per-scene optimized methods, our method can obtain comparable performance (93.3% v.s. 93.5% 2D mIoU) on novel views. Moreover, only our methods can be generalized to novel scenes (47% 2D mIoU). Although Semantic-NeRF performs better than our method (+3.5%), it requires per-scene optimization and thus not generalizable to unseen scenes. Comparing with other generalizable methods, our method obtain the state-of-the-art performance. For example, our method significantly improves over Mask2Former by 7%, which shows our method is better and more concise. Furthermore, our improves significantly (71.6% v.s. 56%) compared with Sematic-Ray. In addition, our method has improved tremendously (71.6% v.s. 39.1%) over the naive approach that incorporates a semantic head into the generalizable NeRF framework. This validates the effectiveness of our soft voting scheme.

Table 2: Quantitative comparison on 3D semantic segmentation from the val and test set of ScanNet. Note that 'Geometry' means using 3D geometry as input and 'Sup.' represents supervision.

|  | Method | Val Split (mIoU) | Test Split (mIoU) | Geometry |
|---|---|---|---|---|
| 3D Sup. | Atlas [31] | 36.8 | 34 | ✗ |
|  | NeuralRecon [40] | 37.5 | 35.1 | ✗ |
|  | Joint Recon-Segment [29] | - | **51.5** | ✗ |
|  | PointNet++ [33] | 53.5 | 33.9 | ✓ |
|  | PointConv [47] | **61** | 55.6 | ✓ |
| 2D Sup. | S4R [20] | 36.9 | - | ✓ |
|  | ScanComplete [15] | 29.5 | - | ✓ |
|  | Ours | 58.8 | 55.1 | ✗ |
|  | Ours with Mesh | **60.4** | **55.8** | ✓ |

### 5.1.2 3D semantic segmentation

We evaluate 3D semantic segmentation on the val and test set of ScanNet. We compare our method with approaches trained on 3D semantic supervision, *i.e.*, Atlas [31], NeuralRecon [40], Joint Recon-Segment [29], PointConv [47], and PointNet++ [33], and approaches trained on 2D semantic supervision, *i.e.*, S4R [20], and ScanComplete [15]. The input of these methods are different: PointConv [47], PointNet++ [33], S4R [20], and ScanComplete [15] use 3D geometry, e.g., point cloud, 3D mesh, etc, as input; Others, *i.e.*, Atlasm NeuralRecon, and Joint Recon-Segment [29] use RGB images as input to predict 3D surfaces and use an extra semantic head to predict semantics. In Tab. 2, we use 'Geometry' to denote 3D geometry as input and 'Sup.' to represent supervision.

As shown in Tab. 2 , our proposed method achieves comparable performance to the methods with 3D semantic supervision, and significantly outperforms all methods with 2D semantic supervision. For example, our approach performs significantly better than all other methods using RGB images as

Table 3: Quantitative comparison on Replica.

| Method | Atlas | NeuralRecon | Ours |
|---|---|---|---|
| 3D mIoU | 25.4 | 26.2 | 43.8 |

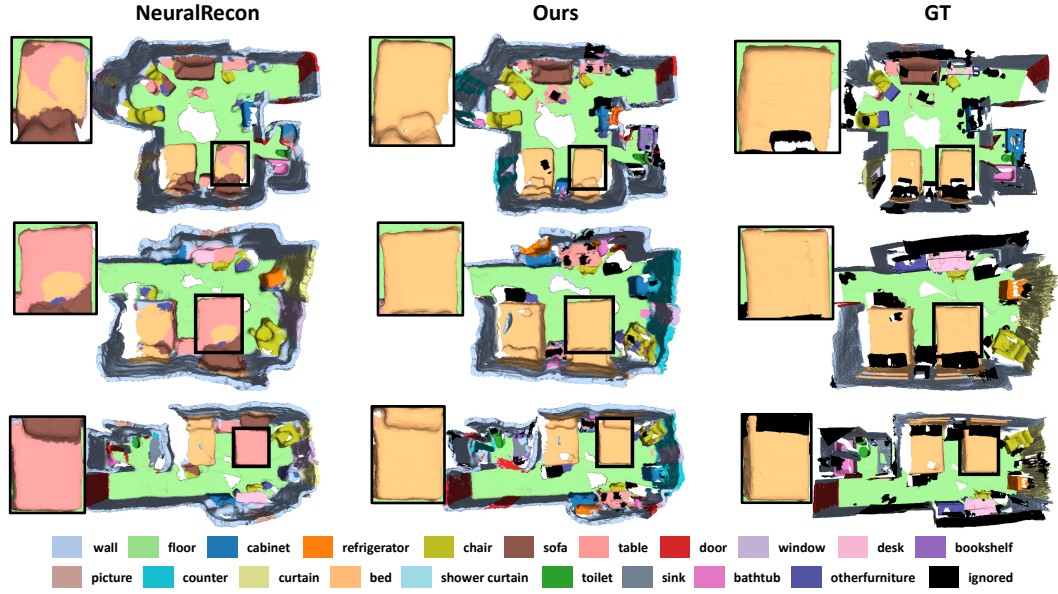

| wall | floor | cabinet | refrigerator | chair | sofa | table | door | window | desk | bookshelf |
| picture | counter | curtain | bed | shower curtain | toilet | sink | bathtub | otherfurniture | ignored |

Figure 3: Qualitative 3D semantic segmentation results on ScanNet. As we can see from the figure, our method is better than NeuralRecon, especially on the 'bed' class. In detail, NeuralRecon mistakenly classifies the bed as a table (highlight in black box).

input. The highest mIoU of the previously best method that predicts 3D surface on the ScanNet test set is $51.5\%$ [29], which is significantly less than our results of $55.1\%$. More importantly, they need 3D semantic supervision while we only require 2D semantic supervision. Our approach still performs better than some of the methods ($58.8\%$ v.s. $53.5\%$ [33]) with 3D supervision and 3D geometry as input. Our method is comparable to PointConv which uses 3D Geometry as input ($60.4\%$ v.s. $61\%$) when we replace SDF prediction with ground truth. Similar to our method, S4R leverages differentiable rendering as training supervision. However, S4R incorporates a semantic head into the existing NeRF framework and uses 3D geometry as input. Our approach outperforming S4R ($36.9\%$ v.s. $58.8\%$ ) proves the effectiveness of our proposed soft voting scheme. We also show qualitative comparison with NeuralRecon in Fig. 3. We can see that NeuralRecon wrongly classifies the bed as table (highlight in black box), while our approach makes the correct prediction. Tab. 3 shows the performance of our method compared with Altas [31] and NeuralRecon [40] that also use color images as input when trained on ScanNet and test on Replica. We can see that our method shows much stronger ability ($43.8\%$ v.s. $25.4\%/26.2\%$) for cross-dataset generalization.

## 5.2 Ablation Studies

As shown in Tab. 4, we do ablation study to investigate the contribution of each component proposed in our method on ScanNet. We start with a baseline method predicting the semantics directly of each 3D query point. Next, we use only semantic logits to vote for semantics of each 3D query point. We then performed a series of tests where we include each component of our framework one-by-one and measured the impact on the performance. The third row shows the impact of using a probability distribution instead of semantic logits, and the fourth shows the impact of adding the visibility module. We find that each component improves the 3D segmentation IoU performance.

Table 4: Ablations of our design choices on 3D semantic segmentation.

| Predict Directly | Logit | Prob. | Visibility Module | mIoU | mAcc |
|:---:|:---:|:---:|:---:|:---:|:---:|
| ✓ | ✗ | ✗ | ✗ | 37.5 (+0) | 47.0 (+0) |
| ✗ | ✓ | ✗ | ✗ | 57.3 (+19.8) | 70.9 (+23.9) |
| ✗ | ✗ | ✓ | ✗ | 57.9 (+0.6) | 71.2 (+0.3) |
| ✗ | ✗ | ✓ | ✓ | 58.8 (+0.9) | 72.1 (+0.9) |

# 6  Conclusion

In this paper, we introduce a *generalizable* 3D segmentation framework based on implicit representation. Specifically, we propose a novel soft voting mechanism on 2D semantic information from multiple viewpoints to predict the semantics for each 3D point. Instead of using positional information, our framework utilizes multi-view image features as input and thus avoids the need to memorize scene-specific geometric and semantic details. Additionally, our framework encodes view differences to give higher importance to nearby views compared to distant ones when predicting the voting scores. We also design a visibility module to identify and discard detrimental information from occluded views. By incorporating view difference encoding and visibility prediction, our framework achieves more effective aggregation of the 2D semantic information. Experimental results demonstrate that our approach performs comparably to scene-specific methods. More importantly, our approach even surpasses existing strong supervision-based approaches with just 2D annotations.

## Acknowledgement

This research work is supported by the Agency for Science, Technology and Research (A*STAR) under its MTC Programmatic Funds (Grant No. M23L7b0021).

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
