# GNeSF: Generalizable Neural Semantic Fields Supplementary Material

**Hanlin Chen** [1]    **Chen Li** [1]    **Mengqi Guo** [1]    **Zhiwen Yan** [1]    **Gim Hee Lee** [1]
[1] School of Computing, National University of Singapore
hanlin.chen@comp.nus.edu.sg   gimhee.lee@nus.edu.sg

## 1   Implementation Details

Besides semantic rendering loss, we also have the RGB loss following [16]:

$$\mathcal{L}_r = \sum_{\mathbf{r} \in R} ||\mathbb{C}(\mathbf{r}) - \hat{\mathbf{C}}(\mathbf{r})||_2^2 s, \tag{1}$$

where $\mathbb{C}\hat{(\mathbf{r})}$ is the rendered RGB value, and $\mathbb{C}(\mathbf{r})$ is the corresponding ground truth pixel RGB value. Finally, the total training loss $\mathcal{L}$ is:

$$\mathcal{L} = \mathcal{L}_r + \lambda \mathcal{L}_s, \tag{2}$$

where $\lambda$ is the weight of the semantic loss and is set to 1 to balance the magnitude of both losses.

**Semantic View Synthesis:**   Our method utilizes a collection of source views to generate a desired target view. To extract features from these source views, we employ a network with shared weights. The feature extraction network is built upon a U-Net-based architecture, where the encoder component is the commonly used ResNet34 [3], as implemented in PyTorch [3]. In our implementation, Batch Normalization [4] is substituted with Instance Normalization [12], and max-pooling operations are replaced with strided convolutions. The dimensions of a single input image is $640 \times 480 \times 3$. We train GNeSF about 30 hours on a single RTX3090 GPU with 24GB memory.

**3D Semantic Segmentation:**   We follow the training pipeline in [8, 7, 17] to predict SDF and then predict semantics based on predicted SDF. The image backbone architecture is a customized version of MnasNet [9] initialized with weights pre-trained on ImageNet. Additionally, we incorporate a Feature Pyramid Network [5] within the backbone to extract multi-level features that are more representative. We explicitly construct feature volume grids locally with the resolution $96 \times 96 \times 96$ based on the extracted features of some nearby views, and then fuse it into a global volume, following Neuralrecon [8] and Mononeuralfusion [17]. Each vertex of the feature volume grids is projected to the image feature maps and obtains its image features by interpolation. The features of each vertex are the mean and variance of projected image features across views. Then, the feature volume is processed by the 3D encoder SPVNAS [11] leveraging torchsparse [10] for efficient 3D sparse convolution operations. For semantic field training strategy, we randomly sample a scene and 512 rays as the training batch for each iteration. Furthermore, following NeRF [6], we perform hierarchical volume sampling and simultaneously optimize two networks with identical architecture: a coarse network and a fine network. We fix the number of sample points on a ray to 64 for both coarse and fine sampling. We train our model using Adam optimizer with an initial learning rate of 0.001. GNeSF requires 34 hours for features learning as well as geometry prediction and 7 hours for semantics prediction with 4 RTX3090 GPUs.

37th Conference on Neural Information Processing Systems (NeurIPS 2023).

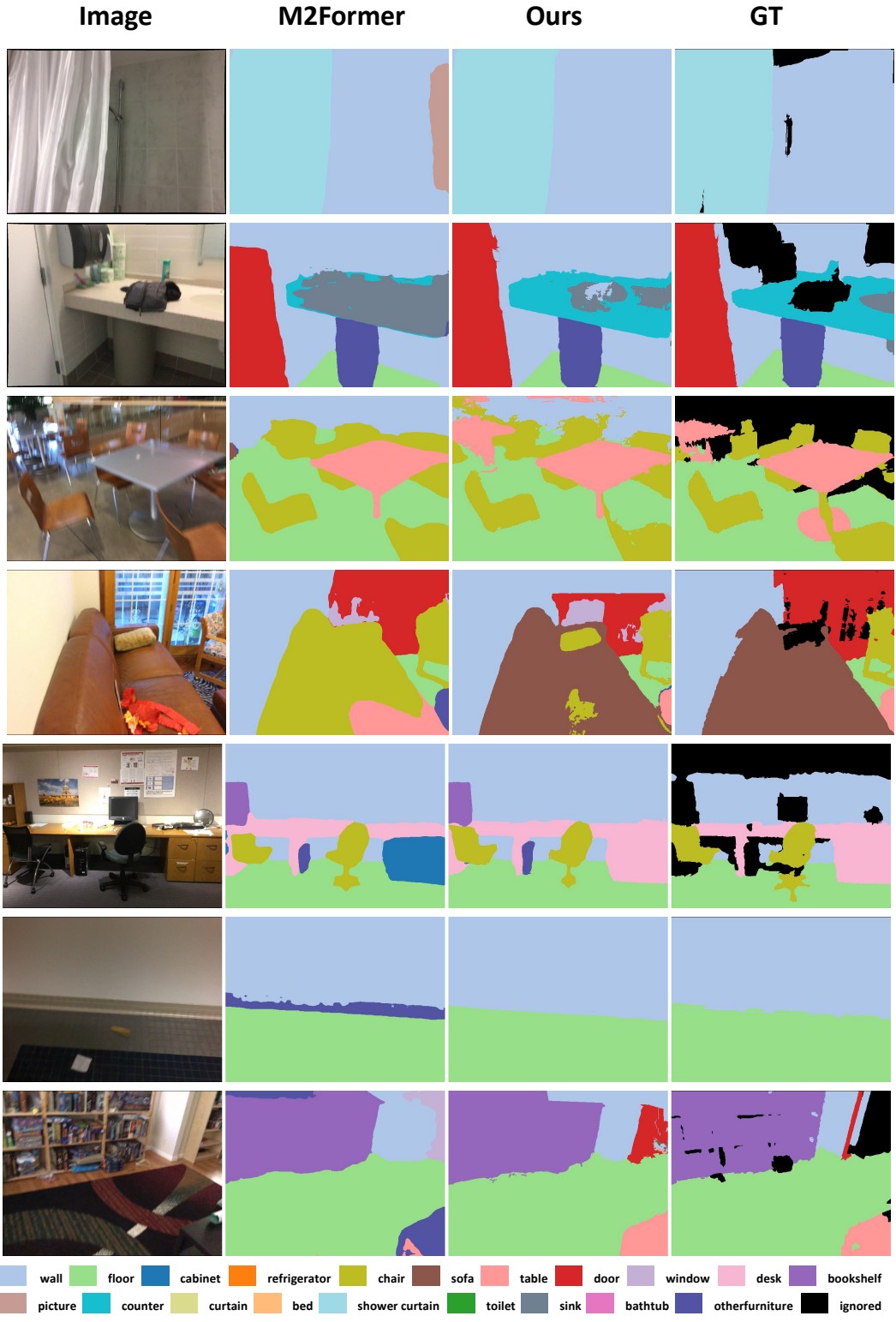

| Image | M2Former | Ours | GT |

wall · floor · cabinet · refrigerator · chair · sofa · table · door · window · desk · bookshelf · picture · counter · curtain · bed · shower curtain · toilet · sink · bathtub · otherfurniture · ignored

Figure 1: Qualitative comparison with Mask2Former for semantic view synthesis on ScanNet. We can see that Mask2Former predicts incorrect segmentation in the the second, fourth, fifth, and last rows. In comparison, our method is able to segment accurately for various scenes.

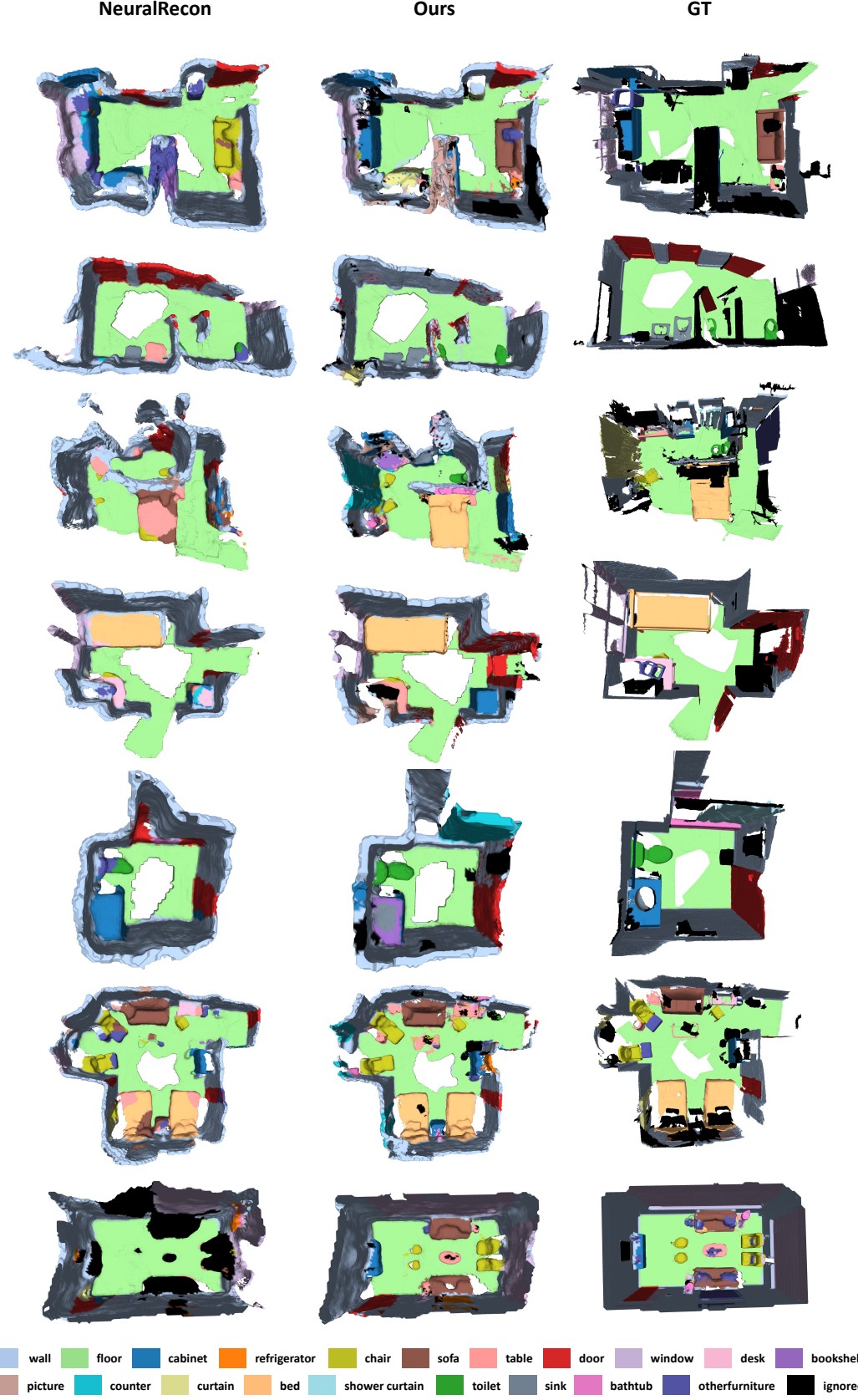

NeuralRecon          Ours          GT

| | wall | | floor | | cabinet | | refrigerator | | chair | | sofa | | table | | door | | window | | desk | | bookshelf |

| | picture | | counter | | curtain | | bed | | shower curtain | | toilet | | sink | | bathtub | | otherfurniture | | ignored |

Figure 2: Qualitative 3D semantic segmentation results on ScanNet (the first row to the sixth row) and Replica (the last row).

## 2 Additional Qualitative Results

### 2.1 Semantic View Synthesis

We show qualitative comparison with another generalizable method Mask2Former [1] in Fig. 1. We can see that our method is better than Mask2Former. In several instances, our method correctly segments objects while Mask2Former produces incorrect results. For example, Mask2Former mistakenly segments the counter as a sink in the second row of Fig.1. Similarly, Mask2Former misclassifies the sofa as a chair in the fourth row, and the desk as a cabinet in the fifth row. Moreover, Mask2Former fails to segment some objects such as the table in the third row. On the other hand, our method successfully segments all objects. Unlike Mask2Former, another advantage of our method is that it avoids excessive segmentation. In the first and sixth row, Mask2Former erroneously segments non-existent elements like the picture and other furniture.

### 2.2 3D Semantic Segmentation

We show more qualitative comparison with NeuralRecon in Fig. 2. We can see that NeuralRecon wrongly classifies the bed as a table, while our approach makes the correct prediction. In addition, NeuralRecon wrongly classifies the sofa as a chair in the first row of Fig. 2. In the second row of Fig. 2, NeuralRecon fails to segment a complete toilet. On Replica (*c.f.* the last row of Fig. 2), our method shows much stronger ability for cross-dataset generalization by correctly segmenting the objects which fails on NeuralRecon.

### 2.3 Color View Synthesis

In Tab. 1, we provide the comparison on ScanNet. The results of other generalizable NeRF methods are from recent SOTA SurfelNeRF [2]. Our method uses the same scene split as SurfelNeRF for training and testing for fair comparison. From the table, we can see that although novel view synthesis is not our focus, our method is still comparable with most generalizable NeRF methods, and worse than the SOTA SurfelNeRF.

Table 1: The comparison of color view synthesis on ScanNet.

| Methods | PSNR↑ | SSIM↑ | LPIPS↓ |
|---|---|---|---|
| IBRNet [13] | 21.19 | 0.786 | 0.358 |
| NeRFusion [15] | 22.99 | 0.838 | 0.335 |
| PointNeRF [14] | 20.47 | 0.642 | 0.544 |
| SurfelNeRF [2] | 23.82 | 0.845 | 0.327 |
| Ours | 22.01 | 0.765 | 0.330 |

## 3 Additional Ablation Studies

**The Effect of the Number of Training Scenes:** We conduct an experimental comparison by setting the number of training scenes to 100, 200, 500, 800, 1000 and 1201 (full training set), respectively. We can see from the Tab. 2 below that the performance is improved when more scenes are used. However, the improvement is marginal when the training scene is increased from 1000 to 1201.

Table 2: The effect of the number of training scenes

| # Scenes | Novel View | | Novel Scene | |
|---|---|---|---|---|
| | mIoU (%) | mAcc (%) | mIoU (%) | mAcc (%) |
| 100 | 93.3 | 96.3 | 47.0 | 60.8 |
| 200 | 92.2 | 95.7 | 58.2 | 69.8 |
| 500 | 91.0 | 94.9 | 65.8 | 76.6 |
| 800 | 89.7 | 94.3 | 68.9 | 79.4 |
| 1000 | 88.6 | 93.7 | 71.2 | 81.5 |
| 1201 (full) | 87.8 | 93.3 | 71.6 | 82.3 |

**Successful and failure cases:** As shown in Fig. 3, we can see that our method can predict correctly in most cases. For example, (a) all 2D semantic predictions are correct; (b) although there is the 2D predictions of a few parts of a object are incorrect, the final 3D predictions of the object are correct. This demonstrate that our proposed soft voting is able to correct the wrong predictions by the pretrained 2D model. However, our method fails in the extreme case when most of 2D predictions across views are incorrect. Although our method is able to predict correct semantics, there are still some wrong parts which manifest themselves in the form of spots as shown in the figure. For example, the red dots in the third row on the chair (c), and the yellow dots in the fourth row on the sofa (d).

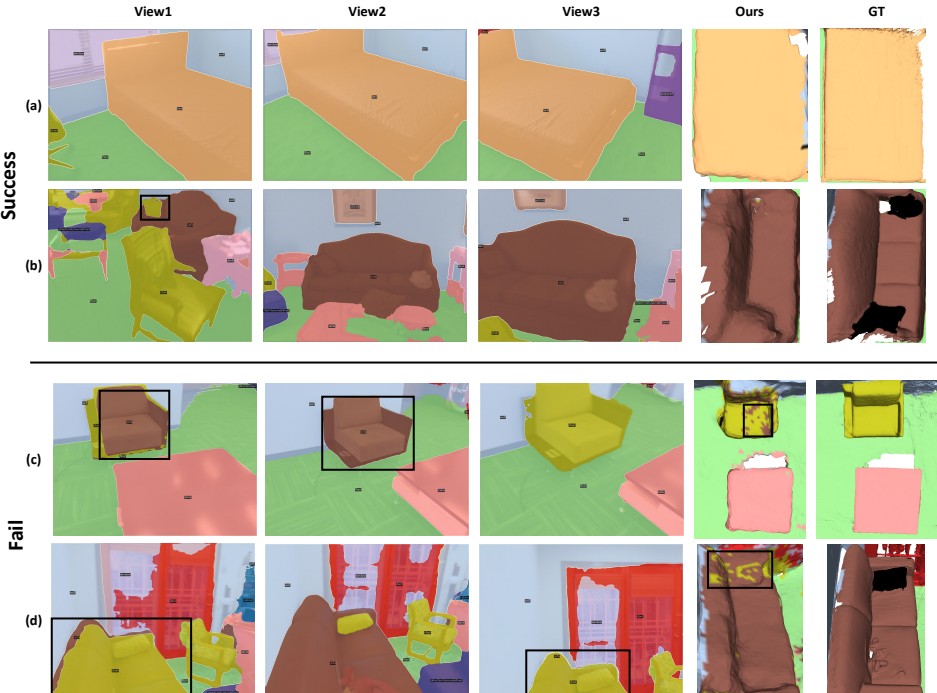

Figure 3: Successful and failure cases of 3D semantic segmentation on ScanNet. We show the 2D semantic prediction of three nearby views by a pre-trained 2D semantic segmentation model (the first three columns), the final 3D semantic segmentation by our (the fourth column), and 3D semantic segmentation ground truth (the last column). The region in 'black box' denotes incorrect prediction.

**The effect of the number of nearby views:** As shown in Tab. 3, we can see that better performance can be obtained with more views. However, the improvement becomes marginal when the number of views is more than 10. Based on this observation, we set the number of nearby views to 10.

Table 3: The effect of nearby views.

| # Views | Novel View | | Novel Scene | |
|---|---|---|---|---|
| | mIoU (%) | mAcc (%) | mIoU (%) | mAcc (%) |
| 5 | 85.4 | 91.2 | 69.7 | 80.3 |
| 8 | 87.4 | 92.9 | 71.4 | 82.1 |
| 10 | 87.8 | 93.3 | 71.6 | 82.3 |
| 12 | 88.0 | 93.5 | 71.7 | 82.5 |

**The performance and speed of GNeSF:** An alternative to GNeSF is training Semantic NeRF with pseudo labels obtained from Mask2Former, namely 'Mask2Former + optimization', so that the Semantic NeRF can infer semantics on novel scenes directly. To compare these two methods, we test both GNeSF and 'Mask2Former + optimization' on 20 scenes randomly selected from 312 validation scenes. The result is shown below in Tab. 4. We can see from the table our method is comparable with 'Mask2Former + optimization' (70.03% v.s. 70.27%). Besides, we compare our

method with 'Mask2Former + optimization' in the aspect of speed, shown below in Tab. 4. Our advantage is efficiency: Semantic NeRF [16] needs 10 hours to finish optimizing a new scene, but our method can do inference directly without further optimization ( 4 minutes). The disadvantage of our method compared to Semantic NeRF is the inference speed (4 minutes v.s. 2 minutes). This is because our method uses many networks such as abundant 2D CNNs while Semantic NeRF is composed of only several fully connected layers.

Table 4: Comparison with 'Mask2Former + NeRF optimization' from several aspects: mIoU, training time, inference time, total time, and speedup. Note that 'Mask2Former + optimization' is abbreviated to 'M2F + Opt.'.

| Method | mIoU (%) | Training (hour) | Inference (minute) | Total time | Speedup |
|---|---|---|---|---|---|
| M2F + Opt. | 70.27 | 10 | 2 | 10+ hours | 1x |
| Ours | 70.03 | 0 | 4 | 4 minutes | 150x |

## 4   Limitation

Although our proposed method benefits from 2D segmentation in most cases, our proposed method cannot work well in extreme cases when 2D segmentation cannot segment objects correctly. Furthermore, the task of the proposed method tries to understand the real world, which could infringe on individuals' privacy rights and raise concerns about surveillance.