# OpenReview forum: "GNeSF: Generalizable Neural Semantic Fields"
_NeurIPS.cc/2023/Conference — NeurIPS 2023 poster_

### Official Review · Reviewer_jaaU · 2023-06-21

**Soundness:** 3 good
**Presentation:** 3 good
**Contribution:** 2 fair
**Rating:** 7
**Confidence:** 4

**Summary:**

The paper combines "generalisable" neural radiance fields and semantic radiance fields. It does so in a relatively straightforward manner, borrowing and combining ideas in these two areas, and contributing a few technical advances on top. It demonstrates good segmentation accuracy both for producing the 2D segmentation of new views and to extract 3D segmentations of the underlying scene geometry.

**Strengths:**

The paper combines the benefits of "generalisable neural fields" and semantic neural fields, essentially inheriting the advantages of both. Compared to prior generalisable neural fields, it adds a semantic dimension, which allows to fuse 2D segmentations into the 3D reconstruction. Compared to semantic neural fields that use test-time optimisation, it can in principle be faster and use a smaller number of images.


**Weaknesses:**

*   Some key aspects of the empirical validation seem to be left untested. The main point of a "generalisable" radiance field compared to a radiance field optimised at test time is to be either (1) faster and/or (2) less data hungry. However, I do not believe the experiments test any of this. Table 1 shows that the proposed "generalisable" field  is comparable (but worse) than semantic NeRF (optimisation based) on the *training scenes*. In a training phase, the "generalisable" field cannot should not provide any advantage in terms of speed or data economy compared to just fitting individual scenes.

    The same Table 1 shows generalisation to new scenes, which would be the main point of the "generalisable" field, but this experiment:
       1. does not compare the accuracy obtained on the new scenes to performing test-time optimisation on the same scenes;
       2. It does not show whether there is a speed advantage compared to test-time optimisation;
       3. It does not show if there is a dat economy advantage compared to test-time optimisation.
    In short, we only know that the generalisable field works to some extend on new scenes, but we have no idea how that compare to test-time optimisation on those scenes, which would be the obvious alternative.

* In Table 1, Ours-100 performs *better* than Ours-1000 (using 10x more data) on the training scenes. This is not entirely unexpected, but it strongly suggests that the "generalisable" neural fields take a significant hit compared to performing per-scene optimisation. This makes it even more important to check this gap on the new scenes. At the very least, the authors should discuss this aspect of the table in some depth.

* line 272: The method applied on new scenes is found to outperform Mask2Former. If I understand the method correctly, thought, it is based on fusing/warping/reweighing the Mask2Former predictions using Eq. (5), even at test time. It is not the model by itself that outperforms Mask2Former, but it is the combination of the model with Mask2Former which is better than Mask2Former alone. It may be a good idea to reinforce this point at line 272.

* The comparison in Table 2 is, for the most part, indicative. It compares different methods which take different kinds of  inputs and are supervised in very different ways. For example, it sounds impressive that this method can be competitive with ScanComplete (2018) that takes as input 3D ground truth, but then again the latter performs segmentation based on an (incomplete) LiDAR point cloud, whereas this method uses Mask2Former segmentations form 2D images of the scene -- hardly comparable.

* It is unclear how the model is trained. The authors do not mention any appearance / RGB loss, so it seems that all the various networks are trained purely with an objective function involving the prediction of segmentation masks. Is this the case, or do you also employ an appearance loss as in most prior "generalisable" radiance fields?

* The goal of the paper, namely combining "generalisable" and semantic neural fields, is a bit incremental. There are elements of novelty in the specific technical details of the combination, but there aren't big surprises.

* Minor issues:
   * line154: the "variance" $\mathbf{v}$ of the feature vectors $\mathbf{F}_i$ should be a matrix, not a vector of the same dimension. Do you mean that this is the diagonal of the covariance matrix?
   * line 156: why do you call $\mathbf{B}$ a *cost volume*? These are primarily just features (the only thing related to the idea of cost is the variance term, the computation of which could be weakly interpreted as comparing features from different views).
   * line 178: $\Delta \mathbf{d}_i$ is a vector, not a distance.



**Questions:**

* Is the "generalisable" radiance field actually faster or less data-hungry than test-time optimisation when applied to novel scenes at test time?

* The "generalisable" radiance field seems to take a significant hit in accuracy when applied to new scenes at test time. How does the accuracy compared to performing test-time optimisation (i.e., semantic NerF) on those?



**Limitations:**

* There is no section dedicated to discuss limitations (even though the latter is recommended in the guidelines). More importantly, I could not find any substantive discussion of the limitations at all.

* There is no discussion of societal impact, but I don't believe this is required for this paper.

---

> ### Author Rebuttal · Authors · 2023-08-09
>
> ## Respose to Reviewer jaaU (R#5)
>
> Thanks for your reviews. We are not sure of the meaning of “test-time optimization” mentioned repeatedly in the reviews. We reckon that this refers to “per-scene optimization” and provides our clarification based on this understanding.
>
> We would also like to clarify the difference between _per-scene optimized NeSF_ and _generalizable NeSF_. For per-scene optimized NeSF, we need to train a semantic NeRF for several hours on each new scene on training data with fully annotated semantic labels. In contrast, the aim of generalizable NeSF is to learn a semantic NeRF that can perform inference on new scenes _without_ further training. Furthermore, generalizable NeSF does not require training data with fully annotated semantic labels for each new scene. Generalizable NeSF is thus more practical than per-scene optimized NeSF because it can apply to new scenes directly without annotating and training.
>
> **Q1**: Compare the accuracy obtained on the new scenes to performing test-time (per-scene) optimisation on the same scenes.
>
> **A**: In our generalizable NeSF setting, there is no annotated training data for novel scenes.  We therefore do not perform per-scene optimization on novel scenes in Tab. 1 since it would be an unfair comparison. Nonetheless, we note that the performance of per-scene optimized NeSF is similar to Novel View in Tab. 1 (about 94%) when annotated data is available for training on novel scenes.
>
> **Q2**: Is there a speed advantage compared to test-time optimisation?
>
> **A**: Yes. Per-scene optimized NeSF needs to annotate labels and optimize first and then perform inference. The annotation and training cost a lot. For example, ignoring the time of annotation, Semantic NeRF [51] needs 10 hours to finish optimizing on a new scene, but our method can infer directly (~4 minutes). This is a 150x speedup.
>
> **Q3**: Is there a data economy advantage or less data-hungry compared to test-time optimisation?
>
> **A**: Compared with per-scene optimized NeSF that requires training on annotated novel scenes, our method performs inference directly on a new scene _without_ training and _does not_ require any semantic annotations.
>
> **Q4**: Why is Ours-100 better than Ours-1000 on the training scenes?
>
> **A**: Ours-1000 performs worse than Ours-100 (87.8 v.s. 93.3) on novel views, but much better than Ours-100 (87.8 v.s. 93.3) on novel scenes. This means that the model trained with more scenes Ours-1000 has better generalization capacity compared with ours-100. The reason is that it is easier for the model to overfit when training on 100 scenes compared to 1000 scenes. Consequently, Ours-100 performs better on novel view synthesis for training scenes, while performs worse on novel testing scenes. Please be noted that the ‘novel view’ refers to testing on the same scenes as training but different views, and ‘novel scene’ means that testing on different scenes from training.
>
> **Q5**: It is fusing the Mask2Former predictions instead of the model by itself that outperforms Mask2Former.
>
> **A**: We apologize for the confusion, we will clarify it in the final version. Indeed, in Line 272 of our paper, we refer to “our method” as the fusion of the 2D semantics predicted by the pretrained Mask2Former, which is called a soft voting mechanism in the paper. Specifically, our soft voting mechanism remedies inconsistencies across multiple views of 2D semantic segmentation via the learning of proper weights, as shown in Fig. 1 of the attached PDF file in the rebuttal.
>
> **Q6**: The comparison in Table 2 is indicative. It is unnecessary to compare with methods with different inputs and supervisions.
>
> **A**: We respectfully disagree. The comparisons in Tab. 2 are with methods trained on 3D semantic supervision while our method is at a disadvantage of being trained on only 2D semantic supervision. Nonetheless, we are still able to show comparable performance with these methods.
>
> **Q7**: Does this method need RGB loss?
>
> A: We do have the RGB loss, but omitted it due to our focus on the semantic field. We will add the RGB loss into the supplementary of our final paper. More details of the training can be found in R#4Q1.
>
> **Q8**: GNeSF is a bit incremental.
>
> **A**: We respectfully disagree. Unlike other semantic NeRF methods that simply add a semantic head, our main contributions are the novel soft voting scheme and visibility module to learn a generalizable neural semantic field. With the proposed modules, our method obtains SOTA on both semantic view synthesis and 3D semantic segmentation. On 3D semantic segmentation, our approach with only 2D annotations can even outperform existing strong supervision-based approaches.
>
> **Q9**: Line154: the "variance" $v$ of the feature vectors $F_i$ should be a matrix.
>
> **A**: The "variance" $v$ is a vector. It is per-channel variance across views following SparseNeuS [a].
>
> **Q10**: Why do you call $B$ a cost volume?
>
> **A**: We follow SparseNeuS [a] to call $B$ a cost volume. To avoid ambiguity, we will call it feature volume in the final version.
>
> **Q11**: $\Delta d_i$ is a vector, not a distance.
>
> **A**: $\Delta d_i$ is indeed a vector, and we bold it in the article to show that it is a vector. To avoid ambiguity, we will call it displacement in the final version.
>
> **Q12**: GNeSF takes a significant hit in accuracy when applying to new scenes. How about the accuracy compared to performing per-scene optimized NeSF?
>
> **A**: Per-scene optimized NeSF does not have generalization ability and thus cannot infer a new scene without training on annotated data. Refer to Q1 for more details on training on a new scene.
>
> **Q13**: Discuss limitations and societal impact.
>
> **A**: The answers can be found in R#1Q6.
>
> [a] Xiaoxiao Long, etc. SparseNeuS: Fast Generalizable Neural Surface Reconstruction from Sparse Views. ECCV, 2022.

---

> > ### Comment · Reviewer_jaaU · 2023-08-11
> > **Response to rebuttal**
> >
> > TL;DR: I am increasing my score, under the expectation that the authors in the final version will be more explicit about the trade-off between their generalisable NeRF and test-time optimisation (unless I am still misunderstanding something fundamental and this is actually unfair as the authors claim).
> >
> > * Rebuttal "In our generalizable NeSF setting, there is **no annotated training data** for novel scenes. We therefore do not perform per-scene optimization on novel scenes in Tab. 1 since it would be an unfair comparison. Nonetheless, we note that the performance of per-scene optimized NeSF is similar to Novel View in Tab. 1 (about 94%) when annotated data is available for training on novel scenes."
> >
> > What do you mean that there are no (2D) annotations at test time? Annotations in this paper are obtained automatically using Mask2Former, so, unless I am missing something fundamental, you absolutely can run Mask2Former at test time and then do test-time per-scene NeRF optimisation on top (basically, Semantic NeRF).
> >
> > If the above is correct, then your method becomes an alternative to test-time 2D semantic segmentation + optimisation, and you should measure both the advantages and disadvantages compared to the latter. Leaving the upper-right corner of Table 1 blank omits a  key data point to assess it.
> >
> > Based on your response, I can surmise that switching to using their generalisable NeRF amounts to a drop of about 25-50 mIoU points (depending on the specific version Ours-100 or Ours-1000) compared to using Mask2Former + test-time NeRF optimisation. So generalisable NeRF is much more efficient (based on the other rebuttal responses, 4 min vs 10 hours, which is a lot!), but also quite a bit less accurate than test-time optimisation.
> >
> > * A: "Ours-1000 performs worse than Ours-100 (87.8 v.s. 93.3) on novel views, but much better than Ours-100 (87.8 v.s. 93.3) on novel scenes. This means that the model trained with more scenes Ours-1000 has better generalization capacity compared with ours-100. The reason is that it is easier for the model to overfit when training on 100 scenes compared to 1000 scenes. Consequently, Ours-100 performs better on novel view synthesis for training scenes, while performs worse on novel testing scenes. Please be noted that the ‘novel view’ refers to testing on the same scenes as training but different views, and ‘novel scene’ means that testing on different scenes from training."
> >
> > Yes, that was my intuition too when reading the paper. My point is that this is an indication that the model starts to underfit the training data at 100 scenes. There is still a test-time generalisation advantage in using more scenes, but the fact that the model underfits during training is another sign that there would be an accuracy gap compared to per-scene test-time optimisation.
> >
> > * A: "We respectfully disagree. Unlike other semantic NeRF methods that simply add a semantic head, our main contributions are the novel soft voting scheme and visibility module to learn a generalizable neural semantic field. With the proposed modules, our method obtains SOTA on both semantic view synthesis and 3D semantic segmentation. On 3D semantic segmentation, our approach with only 2D annotations can even outperform existing strong supervision-based approaches."
> >
> > I recognise that, while the basic problem setup and starting point in formulating a solution a fairly "natural" combination of prior works, making this work in practice is not trivial. When I said that the work feels to me a "bit incremental", I meant that there is good technical progress in the paper, but this progress is not revolutionary.

---

> > > ### Author Response · Authors · 2023-08-15
> > > **Respond to Reviewer jaaU (R#5)**
> > >
> > >
> > > Thank you for affirming our work.
> > >
> > > **Q1**: What do you mean that there are no (2D) annotations at test time?
> > >
> > > **A**: 2D annotation refers to the 2D semantic ground truth labels that are annotated manually. ‘In our generalizable NeSF setting, there is no annotated training data for novel scenes’ –  This means that our generalizable NeSF setting does not require 2D semantic ground truth labels for new scenes. We will make this clearer in the final paper.
> > >
> > > **Q2**: You absolutely can run Mask2Former at test time and then do test-time per-scene NeRF optimisation on top (basically, Semantic NeRF).
> > >
> > > **A:** We agree. Following the suggestion, we train Semantic NeRF with pseudo labels obtained from Mask2Former on 20 scenes randomly selected from 312 validation scenes. To compare with ‘Mask2Former + NeRF optimization’ fairly, our method is also tested on the selected 20 scenes. The result is shown below in Tab. a. We can see from the table our method is comparable with ‘Mask2Former + optimization’ (70.03% v.s. 70.27%). We will add this comparison in the final version.
> > >
> > > **Q3**:  If the above is correct, then your method becomes an alternative to test-time 2D semantic segmentation + optimisation, and you should measure both the advantages and disadvantages compared to the latter.
> > >
> > > **A**: We agree with ‘our method becomes an alternative to test-time 2D semantic segmentation + NeRF optimization’. We compare our method with ‘Mask2Former + optimization’ in two aspects: performance and speed, shown below in Tab. a. Our advantage is efficiency: Semantic NeRF [51] needs 10 hours to finish optimizing a new scene, but our method can do inference directly without further optimization (~4 minutes). The disadvantage of our method compared to Semantic NeRF is the inference speed (4 minutes v.s. 2 minutes). This is because our method uses many networks such as 2D CNNs while Semantic NeRF is composed of only fully connected layers. We will add the advantages and the disadvantages in the final version.
> > >
> > > **Table a**: Comparison with ‘Mask2Former + NeRF optimization’ from several aspects: mIoU, training time, inference time, total time, and speedup. Note that ‘Mask2Former + optimization’ is abbreviated to ‘M2F + Opt.’.
> > > |            | mIoU (%) | Training (hour) | Inference (minute) | Total time | Speedup |
> > > |------------|----------|-----------------|--------------------|------------|---------|
> > > | M2F + Opt. | 70.27    | 10              | 2                  | 10+ hours  | 1x      |
> > > | Ours       | 70.03    | 0               | 4                  | 4 minutes  | 150x    |
> > >
> > >
> > > **Q4**: Leaving the upper-right corner of Table 1 blank omits a key data point to assess it.
> > >
> > > **A**: We will provide the results of using Mask2Former and then test-time NeRF optimization in the final version.
> > >
> > > **Q5**: Switching to using their generalisable NeRF amounts to a drop of about 25-50 mIoU points compared to using Mask2Former + test-time NeRF optimisation.
> > >
> > > **A:** The mIoU is ~94% when test-time NeRF is optimized with ground truth labels. However, the mIoU drops significantly below 94% when test-time NeRF is optimized with pseudo labels generated by Mask2Former.  The mIoU of ‘Mask2Former + test-time NeRF optimization’ shown in Tab. a is a little better than our method (70.27% v.s. 70.03%).

---

> > > > ### Comment · Reviewer_jaaU · 2023-08-17
> > > >
> > > > This further answer clarifies some key points -- particularly the last one:
> > > >
> > > > "The mIoU is ~94% when test-time NeRF is optimized with ground truth labels. However, the mIoU drops significantly below 94% when test-time NeRF is optimized with pseudo labels generated by Mask2Former. The mIoU of ‘Mask2Former + test-time NeRF optimization’ shown in Tab. a is a little better than our method (70.27% v.s. 70.03%)."
> > > >
> > > > I am happy to further raise the score.

---

### Official Review · Reviewer_G1Te · 2023-06-26

**Soundness:** 3 good
**Presentation:** 2 fair
**Contribution:** 3 good
**Rating:** 5
**Confidence:** 4

**Summary:**

This paper introduces a generalizable 3D scene segmentation framework based on neural implicit representation. The proposed framework takes multi-view image features and semantic maps as inputs to reconstruct 3D geometric and semantic information.  The generalizability of the approach enables the synthesis of semantic maps or conducting 3D semantic segmentation for novel scenes with only 2D semantic supervision. Experimental results demonstrate comparable performance with scene-specific approaches and  outperform existing the methods using only 2D annotations.


**Strengths:**

1. This method is novel, and similar approaches have not been proposed in previous papers.
2. This task is useful and also quite innovative.
3. This paper's experiments are comprehensive and thorough.




**Weaknesses:**

1. The article lacks clear explanations for many details, such as the detailed network structure and training strategy. This leads to poor reproducibility.
2. The article claims that only 2D supervision is required, but it does not explain how the SDF (signed distance function) is learned. If following the training pipeline in NeuralRecon or Mononeuralfusion, then this claim of "only needing 2D supervision" is incorrect.


**Questions:**

1. Can the network learn geometry information solely based on equation (6)? If so, why? Is there an intuitive way to explain this?
2. Why does Ours-1000 perform worse than Ours-100?

---

> ### Author Rebuttal · Authors · 2023-08-09
>
> ## Respose to Reviewer G1Te (R#4)
>
> **Q1**: The article lacks clear explanations for many details, such as the detailed network structure and training strategy. This leads to poor reproducibility.
>
> **A**: We will release our source code and trained models after paper acceptance. We have provided some implementation details in supplementary material but will give further explanation here. For the network structure of 3D semantic segmentation shown in Fig. 1, the 2D CNN is MnasNet [a], the 3D encoder is SPVNAS [b], the FC used to predict geometry is one fully connected linear layer, and the FC used to predict weights is three fully connected linear layers. More details about the 2D network and the 3D encoder can be found in NeuralRecon [37]. For semantic field training strategy, we randomly sample a scene and 512 rays as the training batch for each iteration. Furthermore, following NeRF [27], we perform hierarchical volume sampling and simultaneously optimize two networks with identical architecture: a coarse network and a fine network. We fix the number of sample points on a ray to 64 for both coarse and fine sampling. Discussion on the number views can be seen in R#2 Q6.
>
> **Q2**: The article claims that only 2D supervision is required, but it does not explain how the SDF (signed distance function) is learned. If following the training pipeline in NeuralRecon or Mononeuralfusion, then this claim of "only needing 2D supervision" is incorrect.
>
> **A**: By ‘2D supervision’, we refer to 2D semantic supervision since we focus on semantic segmentation. Please be noted that we use’ 2D semantic supervision’ in the abstraction and introduction of our main paper, and only use ‘2D supervision’ in the experiments section occasionally for brevity. We will modify this to avoid confusion in the final version. For learning SDF, we can either use 3D supervision [37] or 2D supervision [52,c,d]. NeuS [c], SparseNeuS [d], and Mononeuralfusion [52] utilize Eikonal loss [e], which regularizes the gradient of SDF to be 1, and a rendering loss to learn SDF. MonoSDF [f] combines both Eikonal loss, and normal and depth predicted from pretrained models to learn SDF with only 2D supervision. Following these methods, it is therefore feasible to implement our method with solely 2D supervision.
>
> **Q3**: Can the network learn geometry information solely based on equation (6)? If so, why? Is there an intuitive way to explain this?
>
> **A**: Yes, we have done this experiment when only using semantic loss, but the network only can learn very coarse geometry information and this will cause the performance to drop. The reason that semantic loss can learn geometry information is that semantic maps can be seen as images with less texture to a certain extent, as shown in Fig. 1 in supplementary material. From the semantic maps in Fig. 1 in supplementary material we can see different objects with outlines, but no details.
>
> **Q4**: Why does Ours-1000 perform worse than Ours-100?
>
> **A**: Ours-1000 performs worse than Ours-100 (87.8 v.s. 93.3) on novel views, but much better than Ours-100 (87.8 v.s. 93.3) on novel scenes. This means that the model trained with more scenes Ours-1000 has better generalization capacity compared with ours-100. The reason is that it is easier for the model to overfit when training on 100 scenes compared to 1000 scenes. Consequently, Ours-100 performs better on novel view synthesis for training scenes, while performs worse on novel testing scenes. Please be noted that the ‘novel view’ refers to testing on the same scenes as training but different views, and ‘novel scene’ means that testing on different scenes from training.
>
> [a] Mingxing Tan, etc. MnasNet: Platform-aware neural architecture search for mobile. In CVPR, 2019.
>
> [b] Haotian Tang, etc. Searching Efficient 3D Architectures with Sparse Point-Voxel Convolution. In ECCV, 2020.
>
> [c] Peng Wang, etc. NeuS: Learning Neural Implicit Surfaces by Volume Rendering for Multi-view Reconstruction. NeurIPS, 2021.
>
> [d] Xiaoxiao Long, etc. SparseNeuS: Fast Generalizable Neural Surface Reconstruction from Sparse Views. ECCV, 2022.
>
> [e] Amos Gropp, etc. Implicit geometric regularization for learning shapes. ICML, 2020.
>
> [f] Zehao Yu, etc. MonoSDF: Exploring Monocular Geometric Cues for Neural Implicit Surface Reconstruction. NeurIPS, 2022.

---

> > ### Comment · Reviewer_G1Te · 2023-08-16
> > **Response to rebuttal**
> >
> > After reading the rebuttal from authors, I decide to keep my score and lean towards positive.

---

### Official Review · Reviewer_1Moh · 2023-07-01

**Soundness:** 3 good
**Presentation:** 2 fair
**Contribution:** 3 good
**Rating:** 4
**Confidence:** 5

**Summary:**

This paper, titled GNeSF, focuses on tackling the challenge of generalizable semantic segmentation using NeRF representation for both novel segmentation synthesis and 3D segmentation. The authors introduce several key contributions in this regard. Firstly, they propose utilizing a cost volume to construct a comprehensive global feature volume. Secondly, a pre-trained 2D segmentor is employed to extract semantic information from the source views. Finally, a soft voting mechanism is introduced to predict the ultimate segmentation label for the target pixel. These approaches collectively enhance the overall effectiveness and robustness of the segmentation process.

**Strengths:**

1. The authors tackle a crucial problem in 3D scene understanding by extending the 2D segmentation map into the 3D space, enabling segmentation for any viewpoint.
2. The framework design presented by the authors appears logical and well-founded. Leveraging the cost volume proves to be an effective approach for encoding comprehensive scene information, while the utilization of a pre-trained 2D model allows for accurate inference of per-view semantics.
3. The introduced voting scheme successfully addresses the challenge of visibility when assigning labels to new pixels. Notably, the experimental results demonstrate a noteworthy improvement compared to previous methods, further validating its efficacy.

**Weaknesses:**

1. Lack of references: The authors utilize the cost volume as a crucial representation for encoding 3D scene information. However, they do not provide explicit details regarding the construction process, nor do they reference important relevant papers in the deep learning era [1][2]. Furthermore, the survey conducted on the "Neural Semantic Field" appears insufficient, as it fails to mention several per-scene segmentation methods based on NeRF representation [3][4][5].

2. Missing technical details: The meaning of "each vertex" mentioned in line 151 is unclear. Did the authors explicitly construct scene volume grids by determining the near and far plane for each view? Additionally, the process of interpolating the semantic logits (line 170) is not adequately explained. Furthermore, it is unclear why the authors utilize the distance between each point x and its camera origin (line 178). Given that this is an important contribution of the proposed method, it is highly recommended to highlight the motivation for using \delta_d at the beginning of this section.

3. Experimental results: It is unclear why the novel view synthesis quality decreases from 93.3 to 87.8 in Table 1 when including more scenes for training. Could you clarify if the term "novel view" refers to evaluating the training scenes while leaving several views for evaluation?

[1] Mvsnet: Depth inference for unstructured multi-view stereo
[2] Cascade Cost Volume for High-Resolution Multi-View Stereo and Stereo Matching
[3] Nerf-sos: Any-view self-supervised object segmentation on complex scenes
[4] Learning Object-Compositional Neural Radiance Field for Editable Scene Rendering
[5] Unsupervised Multi-View Object Segmentation Using Radiance Field Propagation

**Questions:**

The authors primarily focus on showcasing the results of semantic segmentation in their work. However, it is equally important to assess the quality of view synthesis (color image) when rendering novel views. It would be beneficial if the authors could compare their approach with several representative generalizable NeRF methods on the Scannet dataset. I appreciate the experimental setup and would be inclined to give a higher score if the raised concerns are adequately addressed.

**Limitations:**

See the weakness and quations above.

---

> ### Author Rebuttal · Authors · 2023-08-09
>
>
> ## Respose to Reviewer 1Moh (R#3)
>
> **Q1**: Lack of references and provide explicit details regarding the construction process.
>
> **A**: Thanks for your suggestions. We will add all mentioned references into the Related Work Section of the final version. For example, we will insert the following sentences into line 110: ‘NeRF-SOS [a] combines object segmentation and neural radiance fields to segment objects in complex real-world scenes using self-supervised learning. [b] proposes an object- compositional neural radiance field. RFP [c] designs an unsupervised approach to segment objects in 3D during reconstruction using only unlabeled multi-view images of a scene.’ The construction process of the cost volume is that to construct the geometry encoding volume, we project each vertex of the volume to the image feature maps and obtain its image features by interpolation. We then compute the mean and variance of the projected image features from the source views to capture the global information. The mean and variance are concatenated to build a cost volume. More description can be found from Line 151 to 157, and the similar building process of feature/cost volume can be found in [37,52,d]. We will add these in the final version. And  [d] is not cited in our submission, we will add it into our final version.
>
> **Q2**: Missing technical details: The meaning of "each vertex" mentioned in line 151 is unclear. Did the authors explicitly construct scene volume grids by determining the near and far plane for each view?
>
> **A**: We explicitly construct cost volume grids locally with the resolution $96\times96\times96$ based on some nearby views, and then fuse it into a global volume, following Neuralrecon [37] and Mononeuralfusion [52]. Each vertex of the cost volume grids is projected to the image feature maps and obtains its image features by interpolation. The features of each vertex is the mean and variance of projected image features across views. More details can be found in [37,52,d].
>
> **Q3**: Additionally, the process of interpolating the semantic logits (line 170) is not adequately explained.
>
> **A**: The process of interpolating the semantic logits is similar to the process of interpolating the image features. In detail, when applying bilinear interpolation, each target pixel is determined by taking a weighted average of the four nearest pixels from the original semantics map. These four pixels form a square region around the target pixel. The weights are computed based on the relative distances of the target pixel to the four neighboring pixels.
>
> **Q4**: Furthermore, it is unclear why the authors utilize the distance between each point x and its camera origin (line 178). Given that this is an important contribution of the proposed method, it is highly recommended to highlight the motivation for using $\Delta d$ at the beginning of this section.
>
> **A**: Thanks for the suggestion. We will highlight the motivation at the beginning of the section. As shown in Line 182 and 183, the reason for encoding the distance information for voting weights prediction is that nearby views should contribute more compared to the distant ones. In other words, the predicted weight is inversely proportional to distance difference.
>
> **Q5**: Experimental results: It is unclear why the novel view synthesis quality decreases from 93.3 to 87.8 in Table 1 when including more scenes for training. Could you clarify if the term "novel view" refers to evaluating the training scenes while leaving several views for evaluation?
>
> **A**: Ours-1000 performs worse than Ours-100 (87.8 v.s. 93.3) on novel views, but much better than Ours-100 (87.8 v.s. 93.3) on novel scenes. This means that the model trained with more scenes Ours-1000 has better generalization capacity compared with ours-100. The reason is that it is easier for the model to overfit when training on 100 scenes compared to 1000 scenes. Consequently, Ours-100 performs better on novel view synthesis for training scenes, while performs worse on novel testing scenes. Please be noted that the ‘novel view’ refers to testing on the same scenes as training but different views, and ‘novel scene’ means that testing on different scenes from training.
>
> **Q6**: it is equally important to assess the quality of view synthesis (color image) when rendering novel views. It would be beneficial if the authors could compare their approach with several representative generalizable NeRF methods on the Scannet dataset.
>
> **A**: We provide the comparison on ScanNet in the table shown below. The results of other generalizable NeRF methods are from recent SOTA SurfelNeRF [e]. Our method uses the same scene split as SurfelNeRF for training and testing for fair comparison. From the table, we can see that although novel view synthesis is not our focus, our method is still comparable with most generalizable NeRF methods, and worse than the SOTA SurfelNeRF.
>
> |     Methods    | PSNR↑ | SSIM↑ | LPIPS↓ |
> |:--------------:|:-----:|:-----:|:------:|
> |   IBRNet [42]   | 21.19 | 0.786 |  0.358 |
> |  NeRFusion [49] | 22.99 | 0.838 |  0.335 |
> | PointNeRF [f] | 20.47 | 0.642 |  0.544 |
> | SurfelNeRF [e] | 23.82 | 0.845 |  0.327 |
> |      Ours      | 22.01 | 0.765 |  0.319 |
>
>
> [a] Zhiwen Fan, etc. Nerf-sos: Any-view self-supervised object segmentation on complex scenes. ICLR, 2023.
>
> [b] Bangbang Yang, etc. Learning Object-Compositional Neural Radiance Field for Editable Scene Rendering. ICCV, 2021.
>
> [c] Xinhang Liu, etc. Unsupervised Multi-View Object Segmentation Using Radiance Field Propagation. NeurIPS, 2022.
>
> [d] Xiaoxiao Long, etc. SparseNeuS: Fast Generalizable Neural Surface Reconstruction from Sparse Views. ECCV, 2022.
>
> [e] Yiming Gao, etc. SurfelNeRF: Neural Surfel Radiance Fields for Online Photorealistic Reconstruction of Indoor Scenes. CVPR, 2023.
>
> [f] Qiangeng Xu, etc. Point-NeRF: Point-based Neural Radiance Fields. CVPR, 2022.

---

> > ### Comment · Area_Chair_usLm · 2023-08-20
> > **Comment by AC**
> >
> > Dear authors,
> >
> > Thank you for your detailed responses. In my opinion, the authors' response clarifies the questions **Q1** to **Q5**. Please update the final version correspondingly to improve the clarity.
> >
> > Regarding **Q6**, I agree with the authors that novel view appearance synthesis is not the main focus of this paper, and this is clear from the abstract and the introduction. Hence, being comparable with most generalizable NeRF methods is fine. But I have some clarification questions regarding the results: 1) It is interesting that the proposed method has the lowest LPIPS despite not achieving the best PSNR. The SSIM is clearly worse than Sufel-NeRF while the LPIPS is better. What could be the reason? 2) It is also not clear why the proposed method outperforms IBRNet. Could the authors further elaborate on this?
> >
> > @Reviewer 1Moh, please let us know your thoughts if you have further comments.
> >
> > Best,
> > AC

---

> > > ### Author Response · Authors · 2023-08-20
> > > **Respond to AC usLm**
> > >
> > > We will update the final version according to the rebuttal to improve the clarity.
> > >
> > > **Q1**: It is interesting that the proposed method has the lowest LPIPS despite not achieving the best PSNR. The SSIM is clearly worse than Sufel-NeRF while the LPIPS is better. What could be the reason?
> > >
> > > **A:** As shown in Fig. 1 of the paper on LPIPS [a],  PSNR and SSIM are insufficient for assessing high-order image structure (high-frequency components), i.e. they have a preference towards blurry images (low-frequency components). In contrast, LPIPS is designed to be sensitive to high-order image structure (images with high-frequency components), which is closer to human perception. Since semantic maps have clear outlines of objects (i.e. high-frequency components), our method which includes the supervision by semantic maps inherently encourages clear boundaries, i.e. the high-frequency components. Consequently, it is not surprising that there are cases where our method performs well on LPIPS compared to PSNR and SSIM.
> > >
> > > **Q2**: Why the proposed method outperforms IBRNet?
> > >
> > > **A:** This is likely due to having both semantic and RGB image supervision in our method compared to just RGB image supervision in IBRNet. Particularly, the RGB prediction and semantic prediction share the same voting weights in our implementation. The joint semantic and RGB supervisions are complementary in enhancing the learning of the voting weights towards better RGB rendering.
> > >
> > > [a] Richard Zhang, etc.The Unreasonable Effectiveness of Deep Features as a Perceptual Metric. CVPR, 2018.

---

> ### Comment · Reviewer_1Moh · 2023-08-21
> **Thanks for the authors' response.**
>
> I am pleased to have received the comprehensive response from the authors. I acknowledge the strengths of the paper highlighted by fellow reviewers, and I would be inclined to enhance my rating provided the authors can furnish the requested evidence:
> I am particularly interested in observing the view consistency of the rendered RGB-S maps between your method and MaskFormer across multiple new scenes and diverse viewpoints. This is of significance as the Mask2Former has demonstrated an impressive mIoU of 64.6 without any specific design, even surpassing "Ours-100". Given this context, I presume that a key advantage of your method lies in achieving view consistency across different frames when transitioning to new scenes and viewpoints.

---

> > ### Comment · Reviewer_1Moh · 2023-08-21
> > **More responses from the reviewer**
> >
> > Could the paper mainly highlights the results on new scenes on Tab 1, as the merits of generalizable NeRF setting comes from the generalization to new scenes that are unseen in training.

---

> > > ### Author Response · Authors · 2023-08-21
> > >
> > > **Q1:** Could the paper mainly highlights the results on new scenes on Tab 1, as the merits of generalizable NeRF setting comes from the generalization to new scenes that are unseen in training.
> > >
> > > **A:** Sure, we will highlight that the merit of a generalizable NeRF setting comes from the generalization to novel scenes in the final version.

---

> > ### Author Response · Authors · 2023-08-21
> >
> > **Q1**: About ‘The Mask2Former has demonstrated an impressive mIoU of 64.6 without any specific design, even surpassing "Ours-100".’
> >
> > **A:** We would like to clarify that the mIoU of 64.6 from Mask2Former is obtained when Mask2Former is trained on 1,201 scenes as mentioned in Line 245 of our paper. In contrast, “Ours-100” is trained on 100 scenes. It is therefore unfair to make a direct comparison on these two results. It is more rational to compare Mask2Former with “Ours-1000” in Tab. 1 (64.6% v.s. 71.6%).
> >
> > **Q2:** Given this context, I presume that a key advantage of your method lies in achieving view consistency across different frames when transitioning to new scenes and viewpoints.
> >
> > **A:** Our key contribution is **soft voting**. As shown in Fig. 1 (b) of the attached PDF file in the rebuttal, soft voting can help to achieve multi-view consistency. Furthermore, our **distance-based voting** can also help to predict the correct semantics even when multi-view consistency is wrong. For example, in Fig. 1 (c) of the attached PDF file in the rebuttal, although the predictions of View1 and View2 which are farther than View3 are incorrect, our method can give partially correct predictions from the nearest View3 (i.e. distance-based).

---

### Official Review · Reviewer_kUSs · 2023-07-06

**Soundness:** 3 good
**Presentation:** 3 good
**Contribution:** 2 fair
**Rating:** 5
**Confidence:** 4

**Summary:**

This paper proposed GNeSF, a generalizable neural semantic fields by incorporating contextual and semantic prior information via a U-shaped CNN and a pre-trained segmentor. Combing multi-view fused image-level features as well as soft weighting using multi-view softmax probabilities, the overall pipeline could be trained on diverse scenes and then generalize to unseen scenes with promising performance.

**Strengths:**

- This paper proposes a valid extension to enable a generalizable NeSF model by incorporating deep image features as well as semantic priors of multiple RGB images.
- Promising performance on ScanNet and Replica datasets on 2D/3D shows the effectiveness of the methods. Experiments are relative adequate.

**Weaknesses:**

- Contributions and novelties.
  -  As far as I know, Semantic-Ray [1] from CVPR2023 has already made attempt on learning a generalizable semantic NeRF model via attention, which shares similar spirits to GNeSF. What is the main difference?
My main concerns comes from the lack of necessary comparison and discussions to related baselines such as Semantic-Ray [1] and other generalizable NeRF (NeuRay, MVSNeRF) with a semantic head in order to make the distinction clear.

  - Similarly, the discussion of using logits vs. prob. has also been somewhat investigated in PanopticLifting [2] from CVPR2023, thus further weakens the contributions.

- As to numerical experiments, several necessary information is missing in both main paper and supplement and is a bit hard for readers to really judge the strict fairness of comparison.
   - Supervision of using predicted multi-channel prob tensors vs. hard-labels for baselines. If I understand correctly, multi-channel softmax probabilities are predicted per view to serve for soft voting. For baseline methods, are these multi-channel prob tensors also used for training like [2] or only hard discrete labels are used for training baselines which lost categorical information across classes?
  - How many nearby views are used for building the cost volumes and how does the number of views affect the final performance?
  - I also wonder what is the specific dimension/resolution of the built cost volume? As to cover the whole scene, I assume the cost volume owns a large spatial dimension and may make the training costly? Its dimension will affect the represented level of details of deep features.

- Ablations
  - It seems that the mentioned design choices contributes quite marginally to the final performance as shown in Table4
  - In Section 3.2, authors discusses the motivation why soft voting is used instead of directly regressing semantics. It would be good if related results could be shown in ablations (like Table4).


Other minor comments which need further clarify:
 - It would be good to show the training expenses of GNeSF.
- What is exactly the adopted generalizable NeRF in the experiments (i.e., Predict Directly)?
- How to evaluate 3D segmentation as there are discrepancy between reconstructed mesh and GT mesh. Is the same procedure in Atlas used?




[1]Semantic Ray: Learning a Generalizable Semantic Field with Cross-Reprojection Attention CVPR 2023

[2]Panoptic Lifting for 3D Scene Understanding with Neural Fields CVPR 2023

**Questions:**

Please see the comments in the weaknesses section above. My main concern comes from missing comparison to closely related work like Semantic-Ray  and lack explanations of several key components, which makes the technical contribution not clear in my point of view.

**Limitations:**

There is no clear discussion of limitations and potential negative societal impact in the main paper and supplement.

---

> ### Author Rebuttal · Authors · 2023-08-09
>
> ## Respose to Reviewer kUSs (R#2)
>
> **Q1**: What is the main difference with Semantic-Ray [a].
>
> **A**: Although Semantic-Ray [a] and our method both attempt to learn a generalizable semantic NeRF model, the approach taken by the two methods to solve this problem differs significantly. Semantic-Ray proposes a new attention-based feature fusion mechanism for each 3D point and then predicts semantics directly. Specifically, Semantic-Ray fuses features of a 3D point by intra-view radial attention and cross-view sparse attention. Instead of focusing on feature fusion and predicting semantics directly, which is the core contribution of Semantic-Ray, our method focuses on a novel soft voting mechanism to aggregate the 2D semantic information from different views for each 3D point. The advantage of our method is utilizing the SOTA 2D segmentation model to vote for 3D semantics.
>
> **Q2**: Compare with Semantic-Ray and other generalizable NeRF.
>
> **A**: To compare with Semantic-Ray, NeuRay, and MVSNeRF fairly, we train our model with the training scenes from the setting Semantic-Ray and test on the same 10 scenes with Semantic-Ray, shown in the first row of the Tab. 1 of the PDF file in the rebuttal. The results of Semantic-Ray, NeuRay, and MVSNeRF are from the paper of Semantic-Ray. To show the advantage of utilizing abundant 2D data and a large 2D pre-trained model, we use the same setting as our method to train Semantic-Ray on 1201 scenes and test on 312 scenes, shown in the second row of the Tab. 1 of the PDF file in the rebuttal. We can see from the table our method obtains marginal improvement (58.47 v.s. 57.15) when training on 60 scenes compared with S-Ray, but improves a lot (71.62 v.s. 56) when training on 1201 scenes.
>
>
> **Q3**: The discussion of using logits vs. prob. has also been investigated in PanopticLifting [b], weakening the contributions.
>
> **A**: Our core contribution is the proposed novel soft voting mechanism to aggregate the 2D semantic information from different views for each 3D point. We did not claim the use of prob. as our contribution in the main paper. Using prob. is more like a choice of design. Moreover, prob. contributes marginally to final performance as shown in Tab. 4. This shows that our  proposed softing voting method does not critically rely on the prob.
>
> **Q4**: For baseline methods, are multi-channel prob tensors used for training like PanopticLifting [b]?
>
> **A**: For fair comparison, we follow [51, 10] in using hard discrete labels from the ground truth to train all baseline methods and our method. Nonetheless, PanopticLifting [b] utilizes 2D semantic segmentation pretrained on other dataset to generate pseudo labels that are represented as multi-channel prob tensors to optimize the model. This is because PanopticLifting is a per-scene optimized NeSF that cannot infer novel scenes directly without training. The generated pseudo labels are therefore used as surrogates to the ground truth labels that are absent on the novel scenes.
>
> **Q5**: What is the dimension of the cost volume? Does the training cost a lot? The dimension will affect the represented level of details of features.
>
> **A**: We follow Neuralrecon [37] and Mononeuralfusion [52] for constructing the cost volume. Specifically, we first build local cost volumes with resolution $96\times96\times96$, which are then fused into a global one. The local cost volume is processed with 3D sparse convolution and the global cost volume is saved as sparse tensors. Consequently, the training process does not cost much and is similar to [37, 52]. The dimension indeed affects the represented level of details of features because more voxels can be used to represent 3D space, and each voxel can represent smaller structures and details when the dimension is larger.
>
> **Q6**: How many nearby views are used for building cost volumes. How does the number of views affect the final performance?
>
> **A**: $10$ nearby views are used for building the local cost volumes. The effect of the number of views is ablated in the Tab. 2 of the PDF file in the rebuttal. We can see that better performance can be obtained with more views. However, the improvement becomes marginal when the number of views is more than 10. Based on this observation,  we set the number of nearby views to 10.
>
> **Q7**: The mentioned design choices contribute marginally to the final performance as shown in Tab. 4. Tab. 4 should include the result of Predict Directly.
>
> **A**: Tab. 4 does not include our core contribution of soft voting. Instead, we compare soft voting with NeuralRecon that uses directly regressing semantics (Predict Directly) in Tab. 2. We now combine Tab. 2 and 4 in the Tab. 3 of the PDF file in the rebuttal, where we can observe that our proposed soft voting method improves 19.8% mIoU over directly regressing semantics.
>
>
> **Q8**: About the training expenses of GNeSF.
>
> **A**: For Semantic View Synthesis, we train GNeSF about 30 hours on a single RTX3090 GPU with 24GB memory. For 3D semantic segmentation, GNeSF require 34 hours for features learning as well as geometry prediction and 7 hours for semantics prediction with 4 RTX3090 GPUs.
>
> **Q9**: What is the adopted generalizable NeRF as a baseline?
>
> **A**: Our adopted generalizable NeRF baseline is IBRNet [42] with an additional semantic head for semantics.
>
> **Q10**: How to evaluate 3D segmentation?
>
> **A**: We follow the evaluation procedure of Atlas as mentioned in Line 253. The semantic labels from the predicted mesh are transferred onto the ground truth mesh using nearest neighbor lookup on the vertices, and then the standard IoU metric can be used.
>
> **Q11**: Discuss limitations and societal impact.
>
> **A**: The answers can be found in R#1Q6.
>
> [a] Fangfu Liu, etc. Semantic Ray: Learning a Generalizable Semantic Field with Cross-Reprojection Attention. CVPR, 2023.
> [b] Yawar Siddiqui, etc. Panoptic Lifting for 3D Scene Understanding with Neural Fields. CVPR, 2023.

---

> > ### Comment · Reviewer_kUSs · 2023-08-17
> > **Comments on Rebuttal**
> >
> > I appreciate the efforts of authors in clarification towards recent generalizable semantic nerf which addressed some of my concerns.
> >
> > I have a few questions left,
> > 1. As to the numerical comparison to S-Ray, especially on the set-up of training on 1201 scenes and test on 312 scenes, are the set-ups of S-Ray properly adjusted in case the comparison may not be strictly fair? I mean, for example, is the number of trainging steps (and other hyper-params) properly increased since it is initially trained for 260k steps of 60 scenes. Are the same M2F labels used (the same to GNeSF) as training signals for S-ray?
> >
> > 2. I am wondering if the main performance inrease on many unseen scenes comes from the adoptation of using a **cost volume** to explicitly accumalte MVS information (appearance, semantics, geometry, etc) compared to method like S-ray which  purely uses a MLP for representation?
> >
> >     As a consequence, I am not sure  whether the cost volume contributes more than the voting strategy itself.
> >
> >     In addition,  by using such design choice I  find the training expenses greater than methods like S-ray using a single 3090 GPU, though it is not really a weakness of the proposed method.

---

> > > ### Author Response · Authors · 2023-08-19
> > > **Respose to Reviewer kUSs (R#2)**
> > >
> > > **Q1**: Is the number of trainging steps (and other hyper-params) properly increased since it is initially trained for 260k steps of 60 scenes?
> > >
> > > **A:** Yes, we have increased the number of training steps from 260k to 600k. We train Semantic Ray for 600k steps with one single 3090 GPU for about 52 hours. In the table below, we show the effect of training iteration on s-ray. We can see from the table that increasing iterations indeed improve s-ray, but s-ray is still worse than our method (56% v.s. 71.6%).
> > >
> > > |   Method  | S-ray | S-ray| Ours |
> > > |:---------:|:-----:|:----:|:----:|
> > > | Iteration |  260k | 600k | 150k |
> > > |    mIoU   |  52.8 | 56.0 | 71.6 |
> > > |  Time (h) |   22  |  52  |  30  |
> > >
> > >
> > > **Q2:** The training expenses greater than methods like S-ray using a single 3090 GPU, though it is not really a weakness of the proposed method.
> > >
> > > **A:** As shown in the table above, our method is more effective and efficient than S-ray (30 hours v.s. 52 hours, 71.6% v.s. 56%).
> > >
> > > **Q3:** Are the same M2F labels used (the same to GNeSF) as training signals for S-ray?
> > >
> > > **A:** Yes. The training labels of S-ray are the same as M2F and GNeSF. We use ground truth labels to train all these methods.
> > >
> > > **Q4:** If the main performance increase on many unseen scenes comes from the adoptation of using a cost volume to explicitly accumulate MVS information (appearance, semantics, geometry, etc) compared to method like S-ray which purely uses a MLP for representation? Whether the cost volume contributes more than the voting strategy itself.
> > >
> > > **A:** We show the contributions of the cost volume and voting in the table below. “Using cost volume” means we obtain the features of a 3D sample point by interpolating from an explicit cost volume we built. “Without cost volume” means we obtain features of a 3D sample point by: 1) projecting the 3D point to nearby 2D feature maps; 2) obtaining its image features by interpolation; 3) fusing 2D image features from nearby views as 3D features. We follow S-ray in the “without cost volume” setting. “Without voting” means using MLP to predict semantics directly. “With voting” obtains 19.8% improvement on 3D semantic segmentation compared with predicting directly “without voting” (first row v.s. third row). “Using cost volume” obtains minor improvement (1.9%) compared with “without cost volume” (second row v.s. third row). Therefore, the cost volume contributes less than the voting strategy.
> > >
> > > | Cost   Volume   | Voting |     mIoU     |     mAcc     |
> > > |-----------------|:------:|:------------:|:------------:|
> > > |               ✅ | ❎      |   37.5 (+0)  |   47.0 (+0)  |
> > > |               ❎ | ✅      | 55.4 (+17.9) | 69.3 (+22.3) |
> > > |               ✅ | ✅      | 57.3 (+19.8) | 70.9 (+23.9) |

---

> > > > ### Comment · Reviewer_kUSs · 2023-08-19
> > > > **Response from Reviewer kUSs**
> > > >
> > > > Thank authors for giving more implementation details which shows the proposed method outperforms recent generalizable sem-nerf methods and contributions from designed key ingredients .
> > > >
> > > > Please add additional discussions and results in the final version to make the submission stronger and more complete.
> > > >
> > > >
> > > > Overall, I would like to increase my score.

---

### Official Review · Reviewer_xsYz · 2023-07-07

**Soundness:** 3 good
**Presentation:** 3 good
**Contribution:** 3 good
**Rating:** 6
**Confidence:** 4

**Summary:**

This paper proposes a generalizable 3D segmentation framework. The novel components include (a) a soft voting mechanism that incorporates image features, view difference aggregate the 2D semantic information for 2D views to 3D, (b) a visibility module to avoid being affected by occluded views. With only 2D supervision, this method allows for semantic maps synthesis and 3D semantic segmentation for novel 16 scenes. The method is also strongly supported by experiments.

**Strengths:**

1. The ability to generalize is very important in many tasks for efficiency. This paper proposes a generalizable neural semantic field, which shows significant improvements over prior work on unseen scenes.

2. Compared to prior work that uses 3D supervision, this method only needs 2D semantic supervision during training and can be very helpful for 3D segmentation tasks whose annotations are very costly to obtain.

3. Despite being a bit complex, the proposed model includes a few novel and thoughtful components, which may be in helpful to inspire the community.

**Weaknesses:**

1. Not enough intuition is given to explain the qualitative results. Fig. 3 shows that NeuralRecon predicts worse than the proposed method on bed. However, the authors didn't explain why this can happen (which design choice has potentially helped with this)? In the results shown in supplementary Fig. 2, it seems the proposed method has many dotted predictions (eg, red dots on the walls in the second row), what does this mean? Does it mean the proposed method may have some problem predicting the doors on the wall?

2. The clarity of the figures can be improved. For example, "MVS" in Fig. 1 is never explained in the text. The names of the features and neural networks in Fig. 1 sometimes are not exactly the same as the names used in the text, making it a bit hard for the readers to understand.

**Questions:**

Tab. 1 shows that ours-1000 achieves a significant improvement over ours-100 on novel scenes. Have the authors tried on even larger set of training scenes? Could the authors provide some insights on when the marginal gain on expanding the training set will be negligible? How many scenes are Mask2Former and Predict Directly trained on?

**Limitations:**

The authors have not adequately addressed the limitations or potential negative societal impact. The authors could show some failures cases that indicate some common patterns where the proposed method wouldl fail on.

---

> ### Author Rebuttal · Authors · 2023-08-09
>
> ## Respose to Reviewer xsYz (R#1)
>
> **Q1**: Not enough intuition is given to explain the qualitative results. Fig. 3 shows that NeuralRecon predicts worse than the proposed method on bed. However, the authors didn't explain why this can happen (which design choice has potentially helped with this)?
>
> **A**: NeuralRecon does not perform well on ‘bed’ due to the lack of training data for this category as it is on the long-tail end of the scannet dataset. In contrast, our proposed method introduces a soft voting scheme to compute the semantics based on the large pretrained 2D segmentation model. Since the 2D segmentation model is pretrained on large-scale 2D dataset  (e.g. ImageNet21K) that includes abundant data for beds, our proposed method can therefore work well on the ‘bed’ category.
>
> **Q2**: The proposed method has a few dotted predictions, what does this mean?
>
> **A**: The dotted predictions are caused by the inconsistency between nearby views. In most cases, this inconsistency can be remedied by learning proper weights in our soft voting method. However, the dotted phenomenon will appear in some extreme cases where the 2D predictions of most views are inconsistent with each other.
>
> **Q3**: The clarity of the figures can be improved. For example, "MVS" in Fig. 1 is never explained in the text. The names of the features and neural networks in Fig. 1 sometimes are not exactly the same as the names used in the text, making it a bit hard for the readers to understand.
>
> **A**: Thanks for your suggestions. “MVS” means using Multi-View Stereo to build the cost volume. $x^{3d}$ in Fig. 1 is $x$. We will make these clearer in the final version.
>
> **Q4**: Tab. 1 shows that ours-1000 achieves a significant improvement over ours-100 on novel scenes. Have the authors tried on even larger set of training scenes? Could the authors provide some insights on when the marginal gain on expanding the training set will be negligible?
>
> **A**: We did not try on training scenes larger than ScanNet. This is because: 1) We follow the previous works [49,37,51] to use the ScanNet in our experiments for fair comparison. 2) ScanNet is sufficiently large with a training set of 1,201 scenes, a validation set with 312 scenes and a test set with 100 scenes. Nonetheless, to analyze the effect of the number of training scenes, we conduct an experimental comparison by setting the number of training scenes to 100, 200, 500, 800, 1000 and 1201 (full training set), respectively. We can see from the table below that the performance is improved when more scenes are used. However, the improvement is marginal when the training scene is increased from 1000 to 1201.
>
> |   # Scenes  | Novel View |          | Novel Scene |          |
> |:-----------:|:----------:|:--------:|:-----------:|:--------:|
> |             |  mIoU (%)  | mAcc (%) |   mIoU (%)  | mAcc (%) |
> |     100     |    93.3    |   96.3   |     47.0    |   60.8   |
> |     200     |    92.2    |   95.7   |     58.2    |   69.8   |
> |     500     |    91.0    |   94.9   |     65.8    |   76.6   |
> |     800     |    89.7    |   94.3   |     68.9    |   79.4   |
> |     1000    |    88.6    |   93.7   |     71.2    |   81.5   |
> | 1201 (full) |    87.8    |   93.3   |     71.6    |   82.3   |
>
>
>
> **Q5**: How many scenes are Mask2Former and Predict Directly trained on?
>
> **A**: Both Mask2Former and Predict Directly are trained on the 1201 training scenes of ScanNet, which is the same as our method. The reviewer can also refer to line 245 of the main paper for further description.
>
> **Q6**: The authors have not adequately addressed the limitations or potential negative societal impact.
>
> **A**: Although our proposed method benefits from 2D segmentation in most cases, our proposed method cannot work well in extreme cases when 2D segmentation cannot segment objects correctly. Furthermore, the task of the proposed method tries to understand the real world, which could infringe on individuals' privacy rights and raise concerns about surveillance.
>
> **Q7**: The authors could show some failure cases that indicate some common patterns where the proposed method would fail on.
>
> **A**: As discussed in Q2, there are a few dots when the predictions between nearby views from a pre-trained 2D semantic segmentation are inconsistent. Although our method can predict correct semantics, there are still a few wrong parts which manifest themselves in the form of spots as shown in Fig. 1 of the PDF file in the rebuttal. For example, the red dots in the third row on the chair, and the yellow dots in the fourth row on the sofa.

---

> > ### Comment · Reviewer_xsYz · 2023-08-17
> >
> > After reading all reviewers' comments and the responses from the authors, I am leaning towards keeping my original rating.

---

### Author Rebuttal · Authors · 2023-08-09

There are figures and tables in the PDF. In Fig. 1, we show some successful and failure cases and explanations can be found in R#1Q7. Tab. 1 shows the comparison with other generalizable NeSF and explanations can be found in R#2Q2. Tab. 2 shows the effect of the number of nearby views and explanations can be found in R#2Q6. Tab. 3 exhibits the integrated ablations including soft voting and explanations can be found in R#2Q7.

---

### Decision · Program_Chairs · 2023-09-21

**Decision:**

Accept (poster)

**Comment:**

This paper studies generalizable neural semantic fields that allow for 3D semantic prediction on novel scenes without test-time optimization. While the initial reviews are mixed, the authors provide a strong rebuttal that convinces most of the reviewers to be positive. Agreeing with most of the reviewers, the AC believes this work makes good technical progress and is worth publishing.